# Surface energy balance fluxes in a suburban area of Beijing: energy partitioning variability

Junxia Dou[1, 3], Sue Grimmond[2], Shiguang Miao[1, 3], Bei Huang[4], Huimin Lei[4], Mingshui Liao[5]

[1]Institute of Urban Meteorology, China Meteorological Administration, Beijing 100089, China

[2]Department of Meteorology, University of Reading, Reading, RG6 6ET, UK

[3]Key Laboratory of Urban Meteorology, China Meteorological Administration, Beijing 100089, China

[4]Department of Hydraulic Engineering, Tsinghua University, Beijing 100084, China

[5]Miyun Meteorological station, Beijing Meteorological Bureau, Beijing 101599, China

*Correspondence to*: Junxia Dou (jxdou@ium.cn)

**Abstract.** Measurements of radiative and turbulent heat fluxes for 16 months in suburban Miyun with a mix of buildings and agriculture allows the changing role of these fluxes to be assessed. Daytime turbulent latent heat fluxes ($Q_E$) are largest in summer and smaller in winter, consistent with the net all wave radiation ($Q^*$). Whereas, the daytime sensible heat flux ($Q_H$) is greatest in spring but smallest in summer, rather than winter as commonly observed in suburban areas. The results have larger seasonal variability in energy partitioning compared to previous suburban studies. Daytime energy partitioning is between: 0.15-0.57 for $Q_H/Q^*$ (mean summer=0.16, winter=0.46); 0.06-0.56 for $Q_E/Q^*$ (mean summer=0.52, winter=0.10), and 0.26-7.40 for $Q_H/Q_E$ (mean summer=0.32; winter=4.60). Compared to the literature for suburban areas, these are amongst the lowest and highest values. Results indicate that precipitation, irrigation, crop/vegetation growth activity and land use/cover all play critical roles in the energy partitioning. These results will help to enhance our understanding of surface–atmosphere energy exchanges over cities, and are critical to improving and evaluating urban canopy models needed to support integrated urban services, that include urban planning to mitigate the adverse effects of urban climate change.

## 1 Introduction

In the atmospheric boundary layer, turbulent flows are fundamental to mass and energy transport, so are regarded as basic part of comprehensive observation studies of the urban boundary layer, such as the UBL/CLU-ESCOMPTE project in Marseille, France (Mestayer et al., 2005), the BUBBLE project in Basel, Switzerland (Rotach et al., 2005), and the SURF experiment in Beijing, China (Liang et al., 2018).

The eddy covariance method (EC) allows direct measurement of heat and water vapor exchange between the surface and the atmosphere and is considered the best method to obtain turbulent fluxes (Baldocchi, 2003). Since the 1990s, EC methods are gradually undertaken in cities around the world. These measurements have provided information about surface energy exchanges, helped development of parameterizations (Grimmond and Oke, 1999b, 2002; Järvi et al., 2019), and evaluation and application

of land surface models (Grimmond et al., 2010; Järvi et al., 2011; Järvi et al., 2014; Karsisto et al., 2015; Ward et al., 2016; Liu et al., 2017; Kim et al., 2019) and remote sensing products (Kim and Kwon, 2019).

    The EC method sites cover most of the Local Climate Zones (LCZs) (Stewart and Oke, 2012), from compact high-rise, midrise and low-rise (LCZ1-LCZ3), to open high-rise, midrise and low-rise (LCZ4-LCZ6), and large low-rise (LCZ8) (Oke et al., 2017). The land use of these sites are mainly residential

and commercial areas (Vesala et al., 2008; Bergeron and Strachan, 2012; Ao et al., 2016a; Roth et al., 2017; Hong et al., 2020), as well as institutions and universities (Guo et al., 2016), and industrial areas (Grimmond and Oke, 1999b; Offerle et al., 2006b). The ratio of impervious (e.g. buildings, parking lots, roads) to pervious (e.g. trees, grass) land cover in the EC source area span a large range from 22%/78% in a suburban residential site of Łódz´, Poland (Offerle et al., 2006b) to 100%/0 in a car park site of Basel,

Switzerland (Christen and Vogt, 2004). The earliest observations were generally for short periods (few days to several months) (e.g. Grimmond and Oke, 1995; Spronken-Smith, 2002). With more observational experience and recognition of the benefits of longer periods, funders have supported longer measurement periods. Measurements that cover a season, a year, or even many years (e.g. Christen and Vogt, 2004; Moriwaki and Kanda, 2004; Offerle et al., 2006a, b; Vesala et al., 2008; Ward et al., 2013;

Kotthaus and Grimmond, 2014; Ao et al., 2016a; Roth et al., 2017; Tomoya and Masahito, 2017) provide many new insights. Such as at a suburban site in the UK, energy partitioning favors turbulent sensible heat during summer but latent heat in winter and is strongly dependent on land cover fractions (Ward et al., 2013). However, the seasonal variability of energy fluxes normalized by net radiation is relatively small in a residential neighborhood of Singapore, as the measurement site in the equatorial with a very

small variability in the background climate (Roth et al., 2017).

    According to the Local Climate Zones (LCZs) classification (Stewart and Oke, 2012), LCZs 1-4 sites are defined as 'urban' and LCZs 5-9 as 'suburban' sites in this study. If surface is covered by more than 80% vegetation (including trees, grass, crops etc.), this observation station is referred to as rural site.

In Beijing, observations been taken in a densely built-up commercial and residential area (Liu et al., 2012;

Miao et al., 2012, Wang et al., 2015), an open midrise residential - large agricultural area (Dou et al.,

2019), and a rural area (Wang et al., 2015). These include observations for 1-year at the urban site

individually, and at the urban and rural sites concurrently. However, there is still a need to investigate

seasonal variations in the suburban area, to understand the key factors impacting the surface energy

balance at different timescales as to-date only summer-time observations have been analyzed (Dou et al.,

65      2019).

In this study, daily, monthly, and seasonal variations in surface energy fluxes for a suburban site

(Miyun) in the Beijing are analyzed using 16-months of observations. We focus on the impact of site

characteristics and precipitation on energy partitioning.

## 2 Methods

### 2.1 Site description

Miyun, located about 80 km northeast of the center of Beijing city (Tian'anmen Square) (Fig. 1a), has

an area of 50 km$^2$ and population of ~500,000 (in 2019) (Beijing Miyun Statistical Yearbook, 2020). The

Miyun Meteorological Station (MY; 116° 51′ 51″ E, 40° 22′ 39″ N), on the southeast edge of Miyun city

is in a transition zone between the city and the countryside (Fig. 1b, c). Using the Stewart and Oke (2012)

Local Climate Zone classification, the site is LCZ5 - 'Regular Housing (treed/open)' class.

Within a 1 km radius of the Miyun instrumented tower, the surface is 69.6% impervious (20.6%

buildings, 14.6% roads, and 34.3% parking lots and pavement) and 30.3% vegetation (15% wheat/maize

rotation farmland, 7.2% trees, 8.1% grass) based on analysis of a GF-2 High-resolution image (CCRSDA,

2016) with a spatial resolution of 1 m (Fig. 1c). To the east and southeast of the tower, there is farmland.

The mainly residential 6-storey (~18 m) buildings are at varying distances (west: from 170 m; northwest >

200 m; north: > 70 m) of the tower. Other buildings include the farmer's 1-2 storey house (3.5-7.3 m)

which is about 300 m northeast of the tower. Since 2016 to the southwest (500-1000 m), newer taller

(18-34 m) residential buildings have been built. To the south (150-500 m away) are office (height: 8-50

m) and light commercial buildings (small shops, 8-12 m). Southwest of the tower, there is a roundabout

85      that connects the two main roads in the study area: east-west Jingmi Road (170 m to south) and north-

south Tanxi Road (420 m to west).

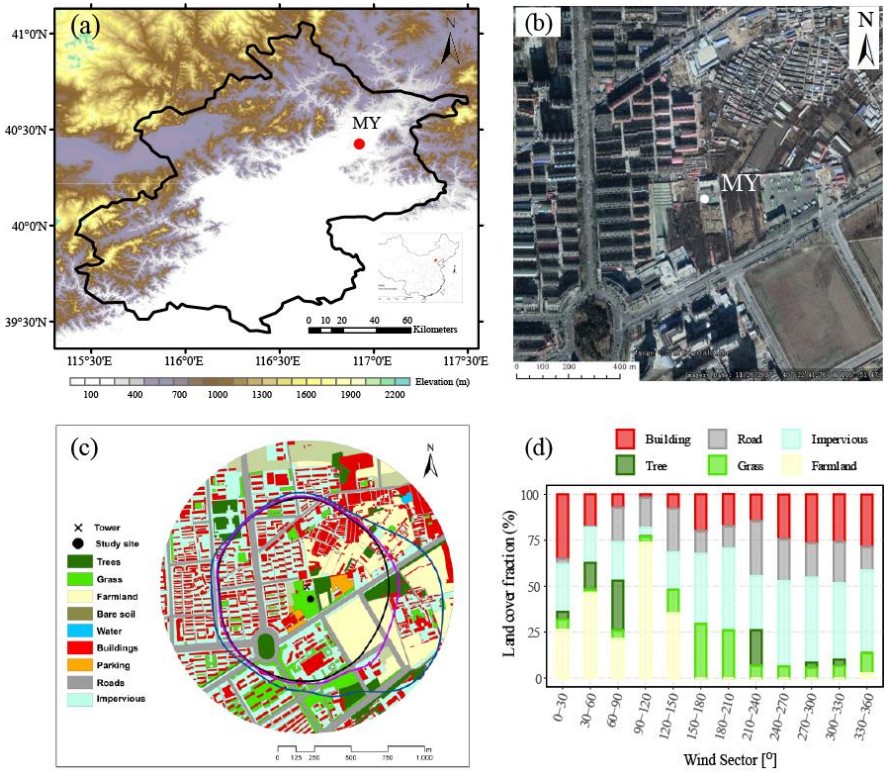

**Figure 1:** Study area (a) Beijing topography with Miyun (MY) (inset; location in China, red dot); (b) aerial view around MY flux tower (Google Earth 2017); (c) land cover within 1 km radius around MY flux tower (black cross) and 90% eddy covariance source area for daytime ($K_\downarrow > 5$ W m$^{-2}$, black line), night (blue) and daily average (pink), and (d) daily mean land cover derived from GF-2 High-resolution image (CCRSDA, 2016) for 30º wind sectors of the 90% source area (shown in c).

**2.2 Instruments and data processing**

Both the eddy covariance (EC) system and radiometer are mounted 36 m above ground level (agl) on a 38-m triangular lattice tower, with the EC system pointing into the prevailing wind direction (east) and the radiometer south to avoid shadows. The EC system's three-dimensional sonic anemometer-thermometer (CSAT3, Campbell Scientific Inc., USA) measures vertical, along-wind, and crosswind velocity and virtual temperature; and the open-path infrared gas analyzer (LI-7500, LI-COR, Inc., USA) measures water vapor and carbon dioxide molar densities. The 10 Hz data are logged on a CR3000 data-logger (Campbell Scientific Inc, USA). The 30 min turbulent sensible and latent heat fluxes are obtained using the EddyPro Advanced (v6.1.0 beta, LI-COR) software with standard correction procedures (Moncrieff et al., 1997) applied to ensure data quality (e.g., de-spiking raw data, tilt correction, time-lag compensation, double coordinate rotation, spectral corrections, and Webb et al. (1980) density

corrections). Any 30-min period with a poor-quality flag (i.e. 2, LI-COR, 2017; N=1552 (6.6%) of $Q_H$,

1987 (8.5%) of $Q_E$) is excluded from this analysis. Data are excluded during rain and 2 h after rain (N=615 (1.4%)). Wind direction data are corrected for changes in magnetic declination as the anemometer was installed with respect to magnetic north.

The radiation fluxes, measured with a CNR4 radiometer (Kipp & Zonen, Netherlands), are 1-min samples by the CR3000 data-logger, from which the 30-min means are calculated. The incoming and outgoing longwave ($L_\downarrow$ and $L_\uparrow$) and shortwave ($K_\downarrow$ and $K_\uparrow$) radiation and net all-wave radiation ($Q^*$) data are restricted to physically reasonable thresholds, with nocturnal shortwave radiation forced to 0 W m$^{-2}$ (Michel et al., 2008).

On the MY tower four levels (10.0, 17.2, 24.2, 36.0 m) of air temperature and relative humidity (HMP45C, Vaisala, Finland) and wind speed (010C, Met One Instruments Inc, USA); and two levels (10.0, 36.0 m) of wind direction (020C, Met One Instruments Inc, USA) are measured.

During the 16 month period analyzed (September 2012–December 2013), after instrument failures (14 January to 18 February 2013) and 91.6% (N=21416, 30 min periods) of the radiation, 78.9% (N=18453) of sensible heat ($Q_H$) and 77.3% (N=18059) of the latent heat fluxes ($Q_E$) data are available.

Additional observations analyzed are from an automatic weather station located 30 m from the EC tower, variables include: air temperature (HMP155, Vaisala, Finland) measured at 1.5 m agl, precipitation from a tipping bucket rain gauge (SL3-1, Shanghai Meteorological Instrument Factory, China) mounted at 0.7 m agl (April to October), and from a weighing bucket rain gauge (DSC1, Aerospace Newsky Technology, China) mounted at 1.5 m agl (November to March); wind speed and direction (ZQZ-TF, Aerospace Newsky Technology, China) measured at 10 m agl, and soil temperature (ZQZ-TW, Aerospace Newsky Technology, China) measured at 0, 0.05, 0.10, 0.20, 0.40, 0.80, 1.6 and 3.2 m below ground. These data are sampled at 1 s, except for wind speed and direction (0.25 s), and averaged to 1 min (WUSH-BH, Aerospace Newsky Technology, China). Hourly means (or accumulated totals for precipitation) are used here. Quality control checks include: plausible range, internal consistency, temporal and spatial consistency, and other standard China Meteorological Administration network checks (Ren et al., 2015).

Soil moisture content is measured from gravimetric samples collected with a straight-shank drill with a scale. Measurements are taken on the 8th, 18th and 28th of each month within the 300 m of the site between 8th March and 28th November each year at five equally spaced depths (0.1 m interval to 0.5 m) in irrigated cropland and for the natural bare soil at five addition levels (i.e. 10 levels of 0.1 to 1 m).

**2.3 Meteorological conditions during the study period**

The study period (1 September 2012 to 31 December 2013) is compared to normal (1991-2020) MY weather station data. The range in mean 2-m air temperature is between -13.3 (23 Dec 2012) and 29.4 °C (17 Aug 2013) at the daily scale, and between -7.0 (Jan 2013) to 26.1 °C (July 2013) for monthly scale (Fig. 2a). There are only small deviations from normal in the monthly mean air temperatures, with biggest differences (December 2012, April 2013) both 3.1 °C cooler than normal.

The study period precipitation differed significantly from normal conditions in the region (Fig. 2b). 2012 was extremely wet, especially on a monthly basis with November (77.4 mm) and December (10.6 mm) about 580% and 400% greater than the Normal (13.4 mm, November; 2.7 mm, December), respectively. In contrast, most of 2013 had below average rainfall, except for June which was wetter (186.7 mm, 223% of the Normal). Notably, dry spells occurred in May, November, and December 2013. Only 0.7 mm (1.6% of Normal) rainfall fell in May and there was no rain in November and December 2013 at all.

Easterly winds (30-120°) prevailed at night and the whole day for every season in MY, with a frequency of 50% and 38% in spring, 54% and 43% in summer, 69% and 54% in autumn, and 72% and 62% in winter, respectively (Fig. 2c-j). During the daytime, wind also came from the southwest direction (180-270°) because of the mountain-valley breeze, despite the southwest wind differed in the start and end times among seasons (Fig. 2g-j). In addition, the strongest winds (wind speed >5 m s$^{-1}$) mainly came from the northeast (30-60°), with a higher relative frequency in spring and winter (Fig. 2c-f).

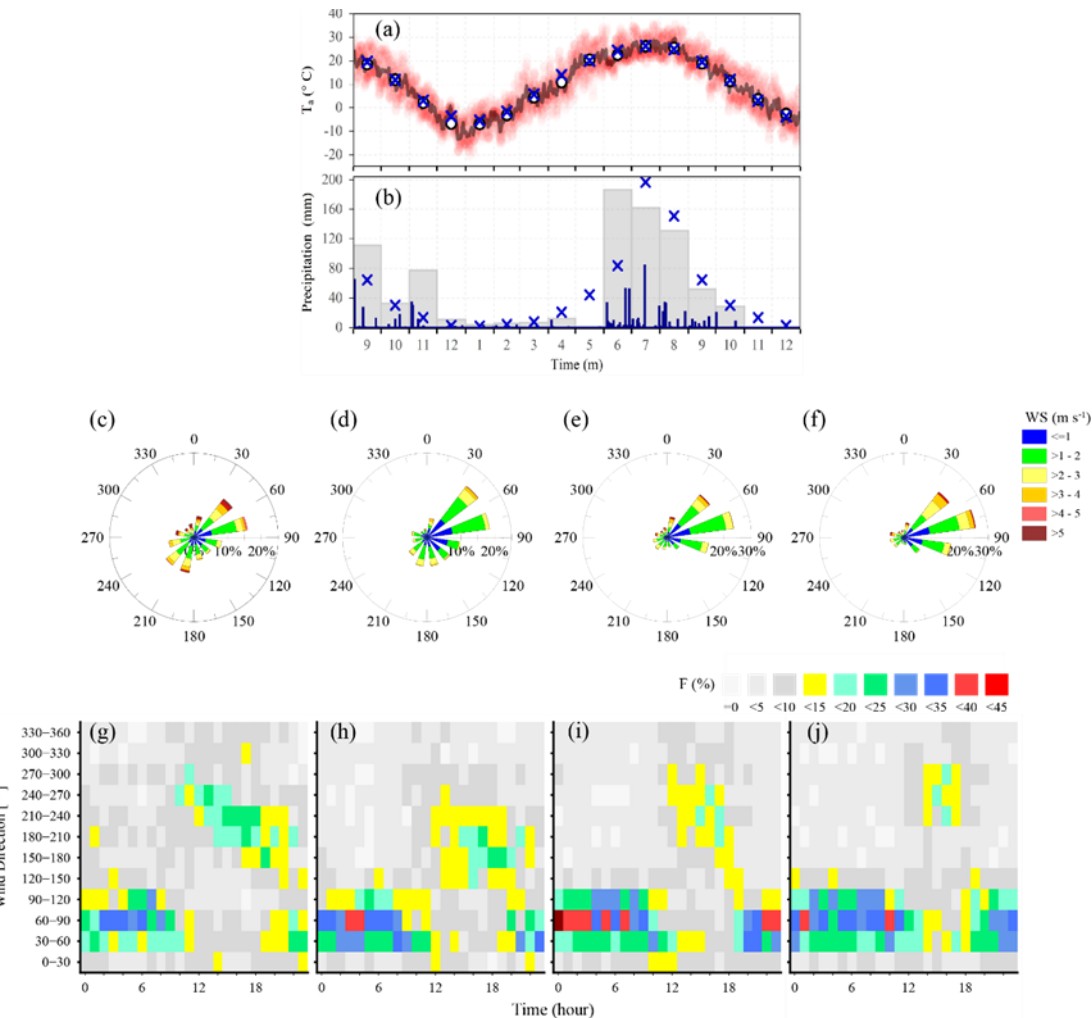


**Figure 2:** Monthly normal (1991-2020, blue crosses) and automated weather station data (September 2012 to December 2013): (a) 2 m air temperature (Ta) as 1 h (dots), daily (solid line), monthly averages (white circles); (b) daily (blue), monthly (grey) precipitation; wind roses (30° bins, 1 h data) stratified by wind speed frequency for (c) spring (MAM), (d) summer (JJA), (e) autumn (SON), (f) winter (DJF); frequency distribution of wind direction (30°, 1 h data) by time of day for (g) MAM, (h) JJA, (i) SON, and (j) DJF.


### 2.4 EC Footprint analyses

To calculate the turbulent fluxes source areas for each 30-min period we use the Kljun et al. (2004) EC footprint model. The roughness length for momentum ($z_0$) and zero plane displacement height ($z_d$) input parameters are based on the height of roughness elements within a 1 km radius of the EC tower. The

buildings vary from 3.0 to 50.4 m. The farmers' houses range from 3.5-7.3 m to the northeast of the tower. The residential buildings (6 floors) are to the west, northwest, and north of the tower, with consistent heights of 16.0-18.5 m. The height of buildings to the south, southwest, and north of the tower vary greatly, but most of the buildings exceed 20 m. The tallest building (50.4 m) is directly 380 m away south

of the tower. The mean weighted by plan area fraction building height ($z_h$) is 13.1 m. The trees are along

roadsides, within roundabout median strip, public gardens and orchards (Fig. 1c). The average tree height

is 8.5 m (trees > 1.5 m), varying between 5.0-21.0 m with tree species differences. The fruit trees in the

orchards (northeast and east of the tower) are a relatively consistent 7.2 to 9.3 m (average =8.6 m). The

mature wheat (June) is 1.1 m and maize (September) is 2.7 m. Using a rule of thumb (Grimmond and

Oke, 1999a) with a mean building height of 13.1 m, $z_0$ is estimated to be 1.3 m and $z_d$ is 9.2 m. Whereas

using Kanda et al. (2013) for 36 -10 degree sectors, give values between 0 and 3.26 m ($z_0$), and from 0

to 36.4 m ($z_d$); and anemometric estimates are 2.9 m ($z_0$) and 4.1 m ($z_d$) (Dou et al., 2019), respectively.

Given that prevailing wind is from the east (30-150º) where vegetation fraction ($\lambda_v$) occupies 60% of the

plan area (≤1 km radius of EC tower), the anemometric values are used. By using the iterative method

suggested by Kent et al. (2017), the impact of these initial values on the probable turbulent flux source

dimensions should become insignificant.

The atmospheric stability parameter ($\zeta = (z-z_d)/L$) is a function of Obukhov length (L) obtained from

the EC observations, the sensor height (z) and $z_d$. Here the variation of turbulence is classified into

unstable as $\zeta < -0.1$, neutral $|\zeta| \leq 0.1$ and stable conditions $\zeta > 0.1$. During the observation period, after

quality control (N=17142, 30 min periods), the stability is predominately unstable (40.9%) and stable

(42.4%), with neutral conditions for 16.7% of the time.

Footprint during unstable and neutral conditions are analyzed together by wind direction (5º sectors)

but split into day ($K_\downarrow > 5$ W m$^{-2}$) and night. The median 90% source area extends to 582-677 m (by

direction) during the daytime and 596-908 m at night (Fig. 1c), which corresponds to a daily median of

559-676 m. Within these source areas, the land use and land cover vary by sector from being more highly

vegetated (30-150º) to more built-up (210-360°) (Fig. 1d).

**2.5 Anthropogenic heat flux**

The LQF version (Lindberg et al., 2018; Gabey et al., 2019) of the large-scale urban consumption of

energy model (LUCY) (Allen et al., 2011; Lindberg et al., 2013) is used to calculate anthropogenic heat

flux ($Q_F$). The temperature response coefficients are calculated for MY using local air temperature and

electricity consumption, vehicle and population data (Appendix A).

**2.6 Storage heat flux**

Given the difficulty of measuring the storage heat flux ($\Delta Q_s$) directly in urban areas (e.g. Offerle et al., 2006b, Roberts et al., 2006), we use two methods to estimate it:

(1) Objective Hysteresis Model $\Delta Q_{s,ohm}$ : see Appendix B for details

(2) Energy balance residual $\Delta Q_{s,res} = (Q^* + Q_F) - (Q_H + Q_E)$ : includes the uncertainty of other terms in the equation. On the one hand, the EC fluxes ($Q_H$ and $Q_E$) underestimate by 10-20% (Wilson et al., 2002; Foken et al., 2008), and on the other hand, given the prevailing easterly winds (30-150º) (Fig. 2c-j), vegetation accounts for 60% of the area of which 44% is cropland (Fig. 1c, d) if $Q_F$ is included, the $\Delta Q_{s,res}$ values should be regarded as the upper limit of heat storage flux (Ward et al., 2013).

**2.7 Data availability and analysis of fluxes**

Analysis is done by season (spring: March–May (MAM), summer: JJA, autumn: SON, and winter: DJF) and by time of day (daytime: $K_\downarrow > 5$ W m$^{-2}$, night: other times).

Missing data are not gap-filled. The mean values of radiation and turbulent fluxes for each half-hour during a day in the month (season) are first calculated to get their mean diurnal patterns. Then the daytime ($K_\downarrow > 5$ W m$^{-2}$) or daily (24 h) mean values are averaged from corresponding periods within the mean diurnal patterns. The daytime (daily) mean ratios are the ratios of daytime (daily) mean values of corresponding radiation and energy fluxes.

**3. Results and discussion**

**3.1 Surface radiation budget**

At the MY site, daytime maxima of incoming short-wave radiation ($K_\downarrow$) range from about 500 W m$^{-2}$ in winter to 1000 W m$^{-2}$ in summer (Fig. 3a). As expected, daytime maxima of $K_\downarrow$ in winter are greater than those for more higher latitude cities (e.g. London, 52 ºN ~ 200 W m$^{-2}$; Kotthaus and Grimmond, 2014) and smaller than for lower latitude cities (e.g. Shanghai, 31.19º N, ~ 600 W m$^{-2}$; Ao et al., 2016b), owing to differences in the solar elevation angle. In summer at MY, the daytime maxima of $K_\downarrow$ is similar to London (Kotthaus and Grimmond, 2014) but slightly higher than for Shanghai (~ 800 W m$^{-2}$; Ao et al., 2016b). This may be attributable to the higher humidity reducing $K_\downarrow$ exceeding the latitudinal differences. The extremely low $K_\downarrow$ values are associated with precipitation, such as recorded in early June 2013 (Fig.

3a). With rain everyday from 4 to 9 June, the $K_\downarrow$ maxima was < 220 W m$^{-2}$ (Fig. 4c-d). The $K_\downarrow$ was 0 W m$^{-2}$ between 11:00-12:00 and 16:00 on 4 June during rainfall (Fig. 4j-k).

The outgoing reflected shortwave radiation ($K_\uparrow$) daytime maxima vary between 65 W m$^{-2}$ (December) and 120 W m$^{-2}$ (July and August). However, some December 2012 and January 2013 exceed 120 W m$^{-2}$ and even 150 W m$^{-2}$ (Fig. 3b) when snow occurred causing albedo becomes an increase to 0.6 and then a decrease with days since snowfall (Fig. 5). When there is no snow on the ground an asymmetry of albedo still exists. With smaller solar elevation, the asymmetry is more pronounced (Figure

Supplementary information S1). Surface heterogeneity is thought to explain this as a basketball court is in the field of view of the radiometer (southeast, Fig. S1a) and albedo will differ from impervious and vegetation surfaces in other parts of the FOV (Fig. S1a), including specular reflection from glass. The building with windows is ~9 m north of the tower, increasing the albedo before noon (cf. afternoon) (Fig. S1b-c). During the spring, summer, and autumn daily midday (10:00-14:00) albedo is between 0.128 and

0.209 in 2012 (0.113-0.209, 2013), while in winter in 2012 (2013) it reaches 0.155-0.478 (0.134-0.293) with the variations in snow cover. Overall, the representative daytime (sunrise to sunset) local-scale albedo is 0.148, which is slightly larger than the midday (10:00-14:00) value (0.145).

      Incoming longwave radiation ($L_\downarrow$) is primarily influenced by near-surface air temperature and water vapor content (Flerchinger et al., 2009; Kotthaus and Grimmond, 2014). Thus, the larger $L_\downarrow$ values are

observed during warm and humid times of the year (i.e. summer - June-August) (Fig. 3c) and smaller values observed in colder winter. The daily maxima of $L_\downarrow$ varies between ~320 W m$^{-2}$ (winter) and ~470 W m$^{-2}$ (summer). The monthly mean $L_\downarrow$ vary between 420 W m$^{-2}$ (July) and 212 W m$^{-2}$ (January 2013). The latter was the coldest month of the observation period (Sect. 2.3).

      Outgoing longwave radiation ($L_\uparrow$) depends on the surface temperature and emissivity. The former

is highly influenced by the total amount of incoming radiative energy. Thus, the larger $L_\uparrow$ is observed in summer (Fig. 3d), consistent with the larger $K_\downarrow$ and $L_\downarrow$. And vice versa, the lowest $L_\uparrow$, along with smaller $K_\downarrow$ and $L_\downarrow$, measured in winter. The highest monthly mean $L_\uparrow$ of 467 W m$^{-2}$ occurred in July and the smallest (275 W m$^{-2}$) again in January 2013.

      As the net all-wave radiation ($Q^*$) is net balance of the four radiation budget components, it is

affected by sun elevation angle, sky conditions, and surface characteristics. The variation of $Q^*$ is similar

to that of the $K_↓$, with daytime maxima of $Q^*$ being greater in summer and less in winter. Daily mean $Q^*$ values vary between -20 W m$^{-2}$ (winter) and 222 W m$^{-2}$ (summer), with large scatter seen from May to September (Fig. 3e). Hence, the largest monthly mean $Q^*$ in summer, with the maximum value (126 W m$^{-2}$) in July 2013. The smaller monthly mean $Q^*$ in June is attributed to its smaller $K_↓$ caused by frequent rainfall. Notably, the 4th June's extremely low $K_↓$ results in a daily mean $Q^*$ than is < 0 W m$^{-2}$ (Fig. 4n). With longer winter nights the monthly mean $Q^*$ is small or even negative. Notably, the minimum value during the observation period is in December 2012 when snow is present (i.e. decreasing $K_↓$) resulting in a monthly mean of only -0.5 W m$^{-2}$ (Fig. 3e).

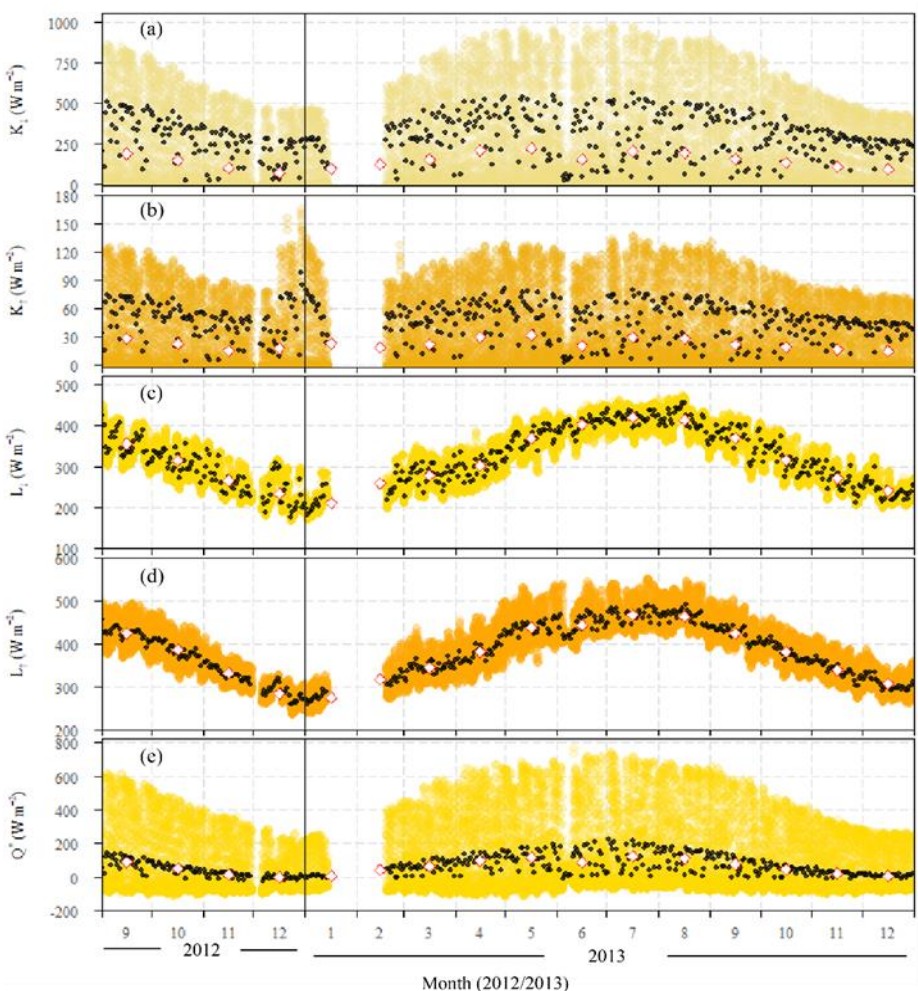

**Figure 3:** Observed 30 min (colour), daily (black) and monthly (white diamonds) means at Miyun (September 2012 to December 2013) of (a) incoming shortwave radiation ($K_↓$), (b) outgoing shortwave radiation ($K_↑$), (c) incoming longwave radiation ($L_↓$), (d) outgoing longwave radiation ($L_↑$), and (e) net all-wave radiation ($Q^*$). Data gap in January and February 2013 due to instrument failure. For the daily fluxes they are daytime ($K_↓$> 5 W m$^{-2}$) for the shortwave fluxes but 24 h for the others.

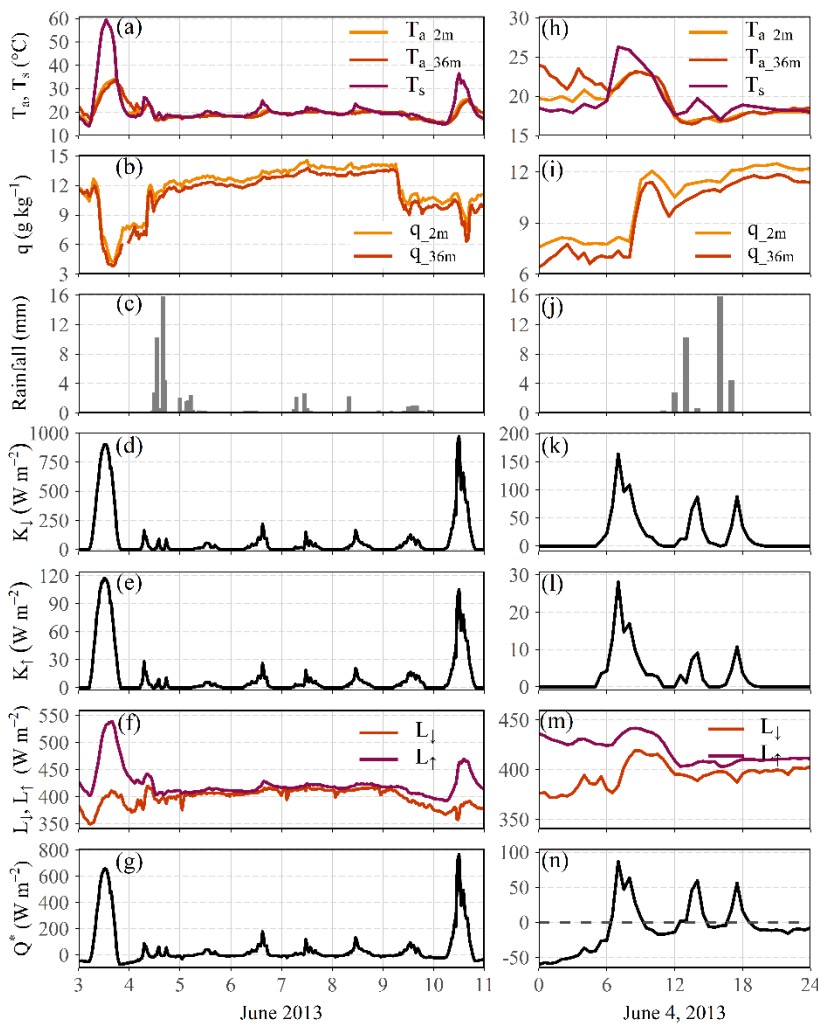

**Figure 4:** Time series of (a, h) air temperature at 2 m and 36 m and surface temperature, (b, i) specific humidity at 2 m and 36 m, (c, j) rainfall, (d, k) incoming shortwave radiation, (e, l) outgoing shortwave radiation, (f, m) incoming and outgoing longwave radiation and (g, n) net radiation for the period (a-g) 3-10 June 2013 and (h-n) June 4, 2013 at Miyun. Rainfall is 1 hour and other data are 30 min resolution.


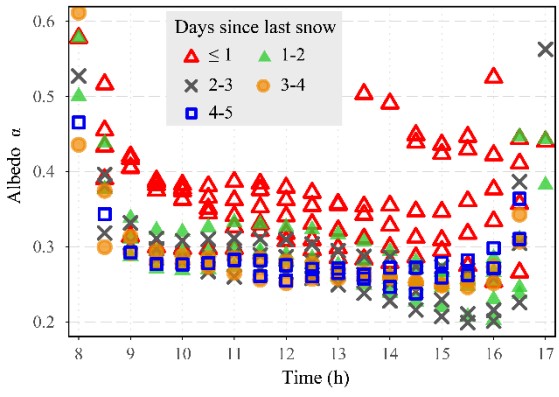

**Figure 5:** Albedo variation with time since last snowfall from December 2012 to February 2013.

**3.2 Anthropogenic heat ($Q_F$) and storage heat ($\Delta Q_S$) fluxes**

The median $Q_F$ vary between 5 and 39 W m$^{-2}$ diurnally across all of the months (Fig. 6). Heat released from buildings ($Q_{F,B}$) dominated $Q_F$, with its diurnal median value ranging from 4.6 to 36.4 W m$^{-2}$ and accounting for 85-95% of $Q_F$ (Fig. S2a). Contribution of vehicle emission ($Q_{F,V}$) is small (1-9% of $Q_F$). The maximum $Q_{F,V}$ value during the morning rush hour (8:00-9:00) is only 1.5 W m$^{-2}$, since the house is usually close to the working place or school and residents do not rely on cars for daily travel or

commuting at MY (Fig. S2b). The diurnal variation of human metabolism ($Q_{F,M}$) is the constant within a year because of the fixed population density (5657 people km$^{-2}$ in 2012 and 5702 people km$^{-2}$ in 2013). Like other studies, $Q_{F,M}$ is small, with values ranging between 0.4-1 W m$^{-2}$ and contributions being 5-8%.

       The fluxes are larger in summer and winter associated with cooling and heating needs, respectively.

Our $Q_{F,B}$ values are larger than estimated in other suburban sites, such as Montreal during winter (Bergeron and Strachan, 2012) and Swindon (Ward et al., 2013), even the air temperature in Montreal is lower in winter. The reason is that Miyun has a greater mean building height (13.1 m) and building cover (21%). Moreover, buildings include hospital and office buildings, which usually have greater energy consumption and emissions than residential buildings. However, our $Q_{F,V}$ values are lower than

those reported in Montreal and Swindon, due to the small number of motor vehicles and various travel modes at Miyun.

       The monthly mean $Q_F$ vary between 17 and 24 W m$^{-2}$ (daily totals = 1.46-2.10 MJ d$^{-1}$ m$^{-2}$). These suburban values are larger than those reported in other suburban residential areas, such as 10-12 W m$^{-2}$ in Montreal during winter (Bergeron and Strachan, 2012), and 6-10 W m$^{-2}$ in Swindon (Ward et al., 2013).

However, these suburban Beijing values are much less than the Beijing city-centre (~130 W m$^{-2}$, Wang et al., 2020). Hence, the MY values appear to be reasonable.

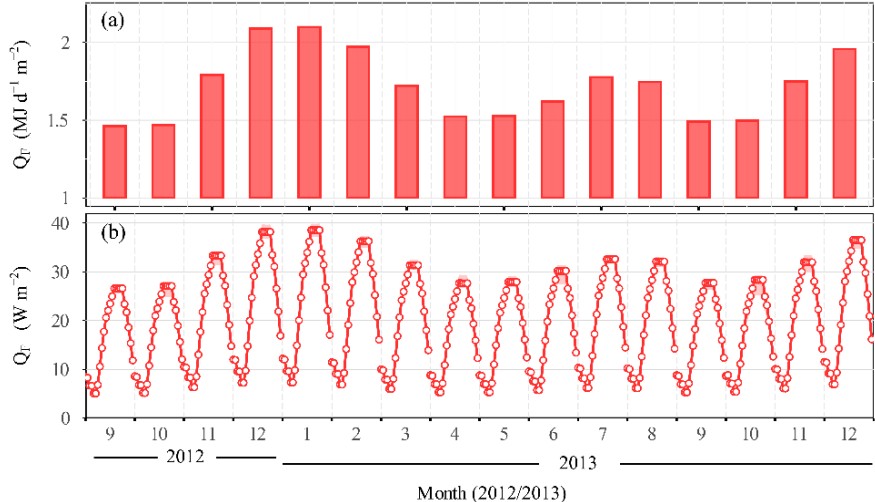

**Figure 6:** Monthly anthropogenic heat flux ($Q_F$) at Miyun (September 2012 to December 2013) calculated as described in Section 2.2: (a) mean total for 24 h, (b) median diurnal patterns with inter-quartile range (IQR) (shading).

The storage heat fluxes are determined using two methods (Sect. 2.6). Generally, storage heat flux is expected to have a net gain in summer and net loss in winter, giving almost zero net heat gain/release over the annual period (Grimmond et al., 1991). From this point of view, $\Delta Q_{s,ohm}$ has a credible annual variation in daily total values (Fig. 7a), but the annual total value of $\Delta Q_{s,ohm}$ is 352.8 MJ m$^{-2}$ in 2013, indicating that $\Delta Q_{s,ohm}$ is overestimated.

In spring and summer, there is little difference in daytime values of $\Delta Q_{s,ohm}$ and $\Delta Q_{s,res}$ except for May 2013 (Fig. 7b), indicating an overestimation of $\Delta Q_{s,ohm}$. The May high $\Delta Q_{s,ohm}$ is the result of both fewer rain events and more radiative input (Sect. 2.3 and 3.1; Fig. 2b, 3a), as it depends on the net radiation value according to the OHM equation. Meanwhile, farmland irrigation prompts more energy to be used for evaporation (Sect. 3.4 and 3.5; Fig. 14, 15), resulting in a decrease in $\Delta Q_{s,res}$. This makes daytime $\Delta Q_{s,ohm}$ obviously higher than $\Delta Q_{s,res}$ in May. In addition, at MY, it is common to see "sunny after rain" in spring and summer, which is similar to the pavement/road watering on sunny days. Pavement heat flux ($G$) is reduced after watering and the maximum value of $G$ is only half of that pavement without watering under the same sunny conditions (Hendel et al., 2015). However, the OHM coefficients of road/pavement/impervious used in this study were fitted from dry surfaces under clear sky conditions (Anandakumar, 1999; Doll et al., 1985; Wang et al., 2008), so it inevitably leads to an overestimation of $\Delta Q_{s,ohm}$ when sunny after rain. The comparison results of MY daytime $\Delta Q_{s,ohm}$ greater than $\Delta Q_{s,res}$ within 24 h after rain in spring and summer confirm this point (Fig. S3).

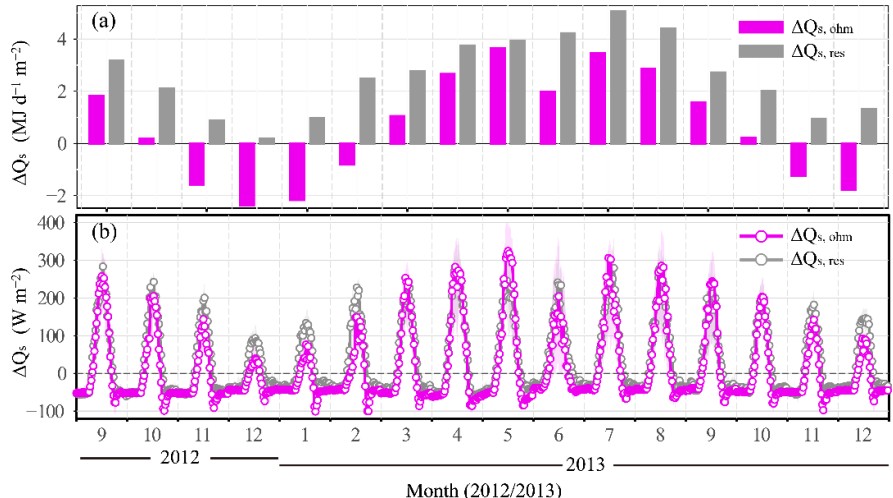

**Figure 7:** Miyun (September 2012 to December 2013) monthly storage heat flux ($\Delta Q_S$) determined using two methods (colour, section 2.5): (a) 24 h mean, and (b) diurnal patterns median with inter-quartile range (shading).

In autumn and winter (November – February), $\Delta Q_{s,ohm}$ is smaller than $\Delta Q_{s,res}$ during the day but larger (more negative) at night (Fig. 7b), similar to residential Swindon results (Ward et al., 2013). As MY site has larger wintertime $Q_F$ emissions, which direct impacts $\Delta Q_{s,res}$ we make a second estimate which omits $Q_F$ and assumes the turbulent heat fluxes are underestimated by 20% (Wilson et al., 2002; Foken et al., 2008) ($\Delta Q_{s,res-v2} = Q^* - 1.2 * (Q_H + Q_E)$). This gives a lower limit, with $\Delta Q_{s,ohm}$ smaller than $\Delta Q_{s,res-v2}$ in winter (Fig. 8). This indicates the winter $\Delta Q_{s,ohm}$ underestimates are related to OHM coefficients for the prevailing wind directions. As cropland covers a greater proportion in the easterly wind direction, it suggests the wheat coefficients are poor in winter (Fig. B2d). This is because smaller fluxes are obtained from using the soil heat flux ($Q_G$) data. These relations with $Q^*$ do not account for those components of heat storage fluxes above the soil heat flux plate, such as ground heat storage and biomass heat storage (Meyers and Hollinger, 2004; Oliphant et al., 2004). They constitute an important proportion of the total storage heat flux. Except for cropland and roof, OHM coefficients of other land cover types are fitted from the spring or summer observation dataset, while these coefficients may not be applicable to autumn and winter and cause the $\Delta Q_{s,ohm}$ to be underestimated in autumn and winter. Long-term datasets covering seasonal changes and surface conditions, especially winter, can provide more OHM coefficients for use (Anandakumar 1999, Ward et al. 2013).

In addition to the OHM coefficients, mismatching the source area between radiation and turbulent fluxes may also cause the uncertainty of $\Delta Q_{s,ohm}$ estimation (Fig. S4). $\Delta Q_{s,res}$ is used in the following

analyses, but it is clearly biased, as monthly daily total values are all positive throughout the year, even

in winter (Fig.7a).

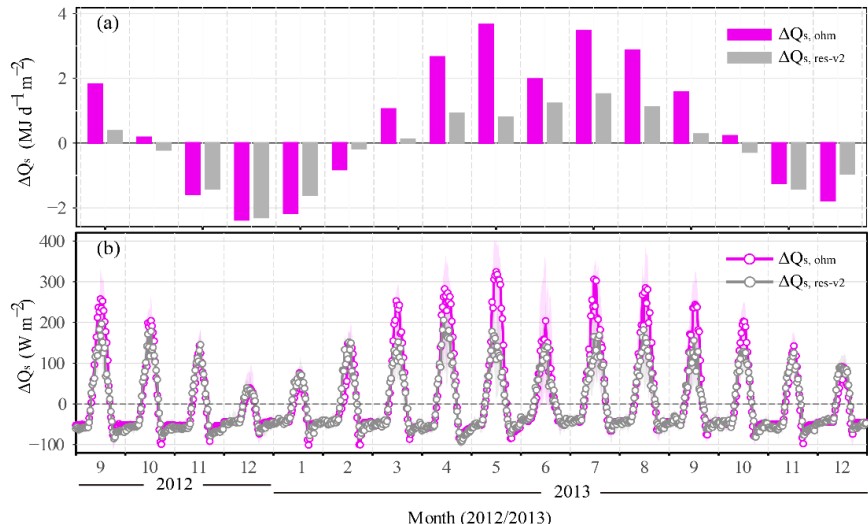

**Figure 8:** Monthly storage heat flux ($\Delta Q_{S,ohm}$ and $\Delta Q_{S,res-v2}$) at Miyun (September 2012 to December 2013) (section 3.2): (a) mean flux for 24 h, and (b) median diurnal patterns with inter-quartile range (IQR) (shading).

**3.3 Monthly diurnal variation of directly observed fluxes**

Typically, $Q^*$, turbulent sensible ($Q_H$) and latent heat fluxes ($Q_E$) each have a unimodal distribution in

the year, but with different seasonal peaks (Fig. 9). Monthly median diurnal maxima of $Q^*$ are higher

from April to September, with the maximum value occurring in August reaching 600 W m$^{-2}$. The daytime

median maximum $Q^*$ (450 W m$^{-2}$) in June is slightly less than other months during this high-frequency

rainfall period. The median diurnal $Q^*$ in December and January is obviously lower than other months,

with a peak value of 180 W m$^{-2}$ in December 2012 being the minimum value during the observation

period. Nighttime $Q^*$ is around -20 W m$^{-2}$ during summer and mostly lies between -80 and -60 W m$^{-2}$ in

other months. Overall, daytime values are greater than nighttime during the investigation period except

for December 2012 (-0.04 MJ m$^{-2}$ d$^{-1}$), which results in monthly daily $Q^*$ being net positive values. These

vary between 0.62 MJ m$^{-2}$ d$^{-1}$ and 10.88 MJ m$^{-2}$ d$^{-1}$ (Fig. 9a, b).

Sensible heat flux $Q_H$ is greatest in spring (March-May) and smallest in June-September 2013 (Fig.

9c, d). The daytime median peak values for these two periods are close to 200 W m$^{-2}$ and ~90 W m$^{-2}$,

respectively. In other months, daytime $Q_H$ maximum values tend to be of between 100-130 W m$^{-2}$, but

in February 2013 approach 170 W m$^{-2}$. In previous sub/urban studies of the East Asian monsoon region,

daytime $Q_H$ values are highest in spring (IAP, Beijing, Miao et al., 2012; Wang et al., 2015; Xianghe, Beijing, Wang et al., 2015; residential area in Seoul, Hong et al., 2020; Seoul Forest Park, Seoul, Lee et al., 2021) or summer (Tokyo, Moriwaki and Kanda, 2004; Shanghai, Ao et al., 2016a; Osaka, Ando and Ueyama, 2017), depending on whether spring is dry with little rain or warm humidity. Meanwhile, daytime $Q_H$ values are lowest in winter when the least solar radiation in the year occurs, except for Xianghe, Beijing and Seoul Forest Park, Seoul. Similar to MY, daytime $Q_H$ values of these two sites are lowest in summer, with irrigated cropland/forest within the flux source area. Hence at MY, the summer daytime $Q_H$ minima is attributed to the extensive irrigated cropland in the prevailing wind direction (Fig. 1c, d; 2d, h). This enhances available energy to support evaporation and transpiration and leads to smaller $Q_H$ values (Dou et al., 2019). Notably, these characteristics are different from wheat/maize rotation farmland under the same climate background. The daytime $Q_H$ values in farmland are lowest in spring, owing to sufficient irrigation and high-water consumption for wheat growth (Fig. S5). Nocturnal $Q_H$ is negative throughout the year, from a mean of -10.5 W m$^{-2}$ in December (2013) to -4.3 W m$^{-2}$ in July, which is similar to those observed at more open suburban sites (Loridan and Grimmond 2012, their Table 2; Oke et al., 2017, their Figure 6.25;). At MY, daily total $Q_H$ on a monthly basis are smallest in winter, 1.07 MJ m$^{-2}$ d$^{-1}$ in December 2013, and largest in spring (4.53 MJ m$^{-2}$ d$^{-1}$, April 2013) (Fig. 9c). Although the daytime $Q_H$ maxima are smallest in summer, the absolute values at night are smaller than that in winter, thus the minimum daily total value occurs in winter.

The seasonal pattern of $Q_E$ is driven by available energy, water supply (including precipitation and irrigation), and phenological variations. Along with the largest $Q^*$ and rainfall, as well as strongest crop growth in summer, daytime $Q_E$ peaks in July, with the maximum reaching 248 W m$^{-2}$ (Fig. 9f). In winter, $Q_E$ is lowest with median diurnal values < 40 W m$^{-2}$. In December 2013, during the long dry spell, with no precipitation from 20 October 2013 (Fig. 2b) and no irrigation (as crops are dormant, typically last irrigation is before 15 November) the maximum value is only 10 W m$^{-2}$. Previous suburban studies report summer and winter daytime $Q_E$ maxima within this range, but at the higher and lower end, respectively (e.g. Grimmond et al., 2002; Spronken-Smith, 2002; Christen and Vogt, 2004; Moriwaki and Kanda, 2004; Goldbach and Kuttler, 2012; Dou et al., 2019). Similar to other urban and suburban studies, throughout the day $Q_E$ is positive (Grimmond and Oke, 1995; Balogun et al., 2009; Ward et al., 2013;

Ao et al., 2016b). At night $Q_E$ is close to zero, being slightly larger during June-August, between 2-19 W m⁻² (mean 7-26 W m⁻² during 20:00-0:00).

Monthly totals of $Q_E$ are between 0.32 and 7.16 MJ d⁻¹ m⁻² (Fig. 9e), with corresponding maximum and minimum monthly totals occur in two seasons (summer - July and winter - December 2013, respectively).

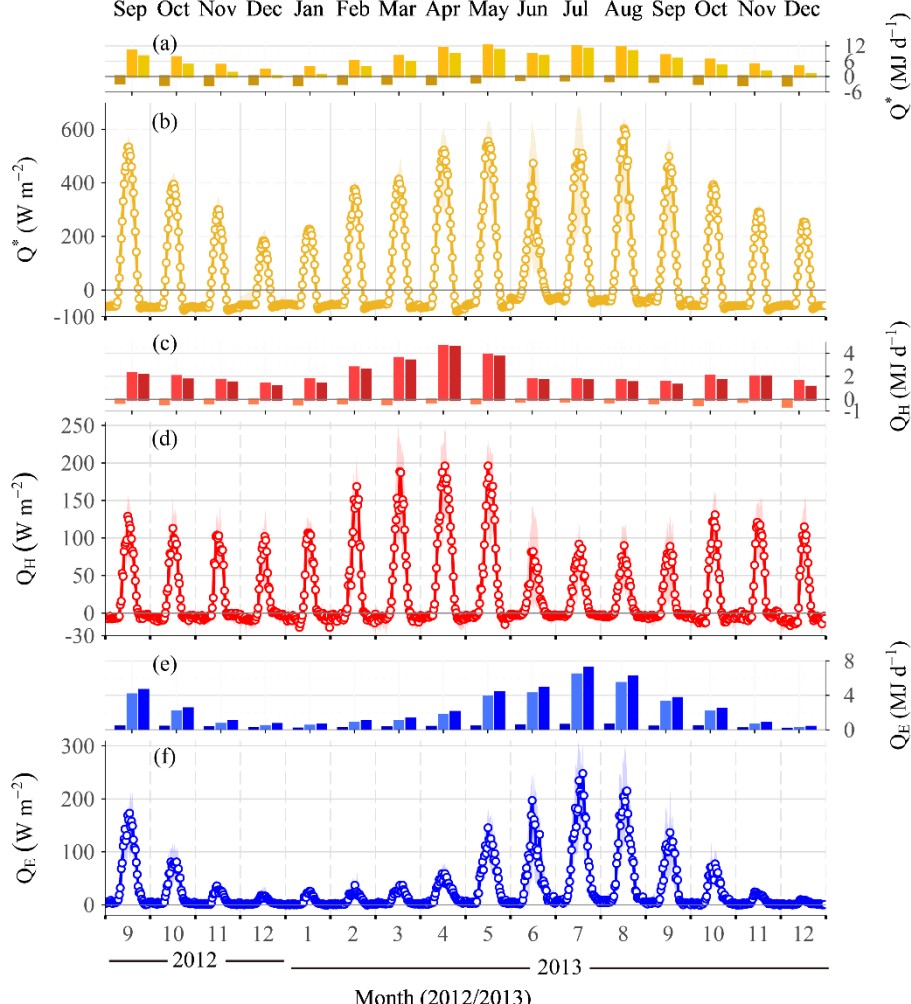

**Figure 9:** Miyun (09/2012-12/2013) monthly fluxes (a, c, e) total energy flux for night, day, and 24 h (left to right) and (b, d, f) diurnal median (line) and inter-quartile range (shading) for: (a, b) net all-wave radiation $Q^*$, (c, d) sensible heat flux $Q_H$, and (e, f) latent heat flux $Q_E$.

### 3.4 Surface energy balance partitioning

To facilitate comparison of surface energy balance (SEB) partitioning between sites a series of average daily, and daytime ($K_\downarrow$ > 5 W m⁻²) flux ratios are analyzed. These include fluxes normalized by net all-wave radiation ($Q^*$) and the Bowen ratio ($\beta = Q_H/Q_E$) (Fig. 10, 11; Table 1). The ratios are calculated at

the 30 min time period allowing the distributions to be analyzed (Fig. 10) but also using monthly mean fluxes.

     During the observation period, daytime $Q_H$ is 15-57% of $Q^*$ (Fig. 10). Unlike the absolute values (Sect. 3.3), $Q_H/Q^*$ is largest in winter and smallest in summer. Whereas $Q_E/Q^*$ has the opposite shape with annual peak in July (Fig. 10). It varies between 6-56% during our study period.

At MY, from November to April (Fig. 10) $Q_H$ dominates the $Q^*$ ratio, whereas from June to September $Q_E$ dominates the $Q^*$ ratio. For May and October, they have a similar ratio with $Q^*$ (Fig. 10). This partitioning pattern is different from that observed in dense residential Tokyo where $Q_H/Q^*$ is always greater than $Q_E/Q^*$ throughout the year (Moriwaki and Kanda, 2004), also unlike the pattern reported in a suburban site Oberhausen where $Q_E$ is almost always dominant (Goldbach and Kuttler,

2012). These pattern differences could be partly attributed to the different vegetation cover, i.e. $\lambda_V$ =34% at MY, 21% at Tokyo, and 61% at Oberhausen.

     The MY June-August daytime $Q_E/Q^*$ ratios between 0.49-0.56 (mean summer =0.52), are greater than most suburban sites, including those with more vegetation coverage than MY, such as Chicago ($Q_E/Q^*$= 0.38; $\lambda_V$ =44%; Grimmond and Oke, 1995), and Basel ($Q_E/Q^*$= 0.30; $\lambda_V$ =53%; Christen

and Vogt, 2004). Therefore, MY June-August daytime $Q_H/Q^*$ ratios (0.15-0.20, summer mean =0.16) are lower in comparison to previous studies (Grimmond and Oke, 2002; Balogun et al., 2009; Goldbach and Kuttler, 2012; Dou et al., 2019). Frequent rainfall and irrigation, prevailing winds from farmland, and rapid crop growth are all believed to enhance the latent heat flux, leading to higher $Q_E/Q^*$ and smaller Bowen ratio values (Sect. 3.5) and consistent with a previous study at the MY (Dou et al., 2019).

In winter, MY daytime $Q_H$ was 41-57% of $Q^*$, which is greater than most suburban sites and some urban sites, including sites with larger plan area of buildings ($\lambda_b$), such as Tokyo ($Q_H/Q^*$=35-40%; $\lambda_b$=33%; Moriwaki and Kanda, 2004), IAP Beijing ($Q_H/Q^*$=28%; $\lambda_b$=68.3%; Miao et al., 2012) and Mexico City ($Q_H/Q^*$=38%; $\lambda_b$=32%; Oke et al., 1999). However, the $Q_E/Q^*$ ratios (0.08-0.18) are comparable to Tokyo ($Q_E/Q^*$= 0.07-0.11; Moriwaki and Kanda, 2004) and slightly larger than that of

IAP Beijing ($Q_E/Q^*$= 0.07; Miao et al., 2012). This suggests the higher winter MY $Q_H/Q^*$ ratio could be attributed to its relatively lower heat absorbed and stored by buildings (here $\Delta Q_s = Q^* - Q_H - Q_E$)

due to smaller $\lambda_b$ (18.9 %). The $\Delta Q_S$ is a small proportion of in the $Q^*$ energy partitioning ($\Delta Q_S/Q^*$= 0.25-0.53).

The partitioning between $Q_H$ and $Q_E$ has clear seasonal variations, with a similar tendency to that

of $Q_H/Q^*$ (Fig. 11 cf. 10). The daytime $\beta$ varies between 0.26 and 7.40 through the year (Fig. 11). This range is much greater than many sites, such as 1.38-5.81 at Tokyo (Moriwaki and Kanda, 2004) and 0.36-1.23 at Oberhausen (Goldbach and Kuttler, 2012). The MY summer and winter (2012/2013) daytime mean $\beta$ (0.32 and 3.08/4.60) are at the lower and higher end of other suburban sites (Fig. 12). Precipitation plays a significant role in the magnitude and amplitude of $\beta$ (see Section 3.5).

At the annual time scale, $Q_H$ and $Q_E$ are 31% and 35% of mean daytime $Q^*$; and 36% and 49% of the daily $Q^*$ (i.e. $Q_E$ dominates) (Table 1). Similarly annual, daytime $\beta$ is 0.89, and daily is 0.73 (i.e. $Q_E$ dominates).

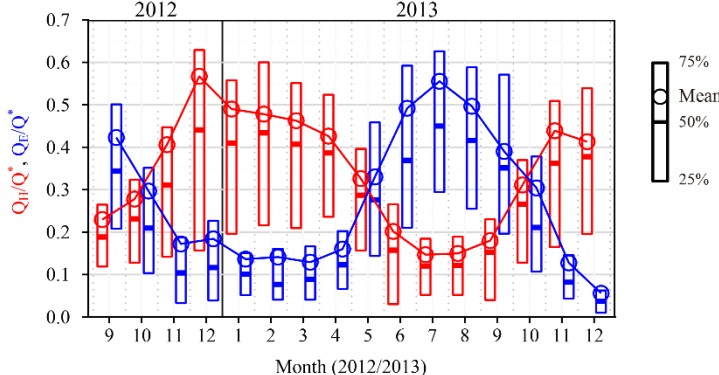

**Figure 10:** Monthly median, IQR and mean (30 min) of daytime ($K_\downarrow > 5$ W m$^{-2}$) sensible heat flux $Q_H$ and latent heat fluxes $Q_E$ normalized by net all-wave radiation ($Q^*$) at MY (September 2012 to December 2013).

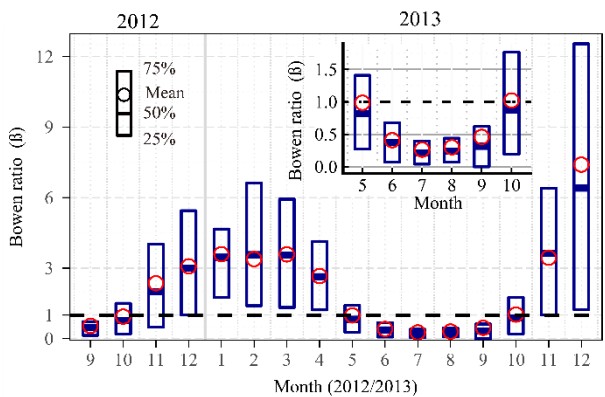


**Figure 11:** As Fig. 7 but Bowen ratio ($\beta$) with inset from May to October 2013.

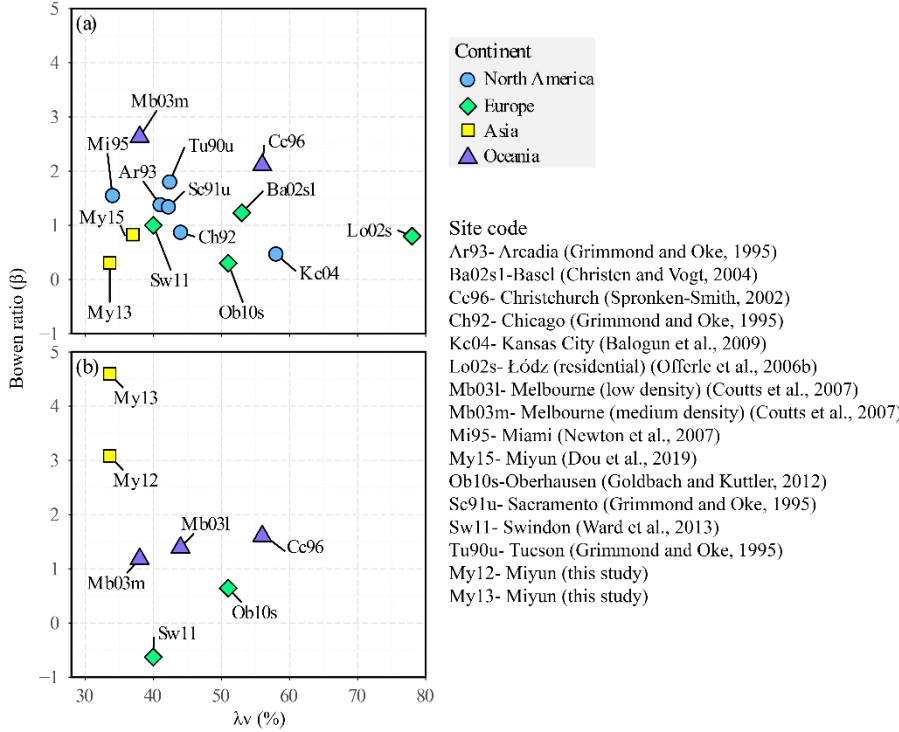

**Figure 12:** Daytime (the definition of daytime is given in the corresponding publication) Bowen ratio during summer (a) and winter (b) versus vegetation fraction ($\lambda_v$) for MY and for various suburban sites in the literature see the legend for references).


**Table 1:** Seasonal and annual mean fluxes and ratios for daytime ($K_\downarrow > 5$ W m$^{-2}$) and daily (24 h) period at Miyun site. Radiation fluxes, net all wave radiation $Q^*$, incoming radiation $Q_\downarrow$ ($Q_\downarrow = K_\downarrow + L_\downarrow$) and incoming and outgoing shortwave radiation $K_\downarrow$, $K_\uparrow$, and incoming and outgoing longwave radiation $L_\downarrow$ and $L_\uparrow$, and albedo= $K_\uparrow/K_\downarrow$; turbulent sensible and latent heat fluxes ($Q_H$, $Q_E$) and their ratios (radiative ratios; and Bowen ratio ($Q_H/Q_E$); and storage heat flux calculated as $\Delta Q_{S,res} = (Q^* + Q_F) - (Q_H + Q_E)$. Anthropogenic heat flux $Q_F$ calculated by LQF version (Gabey et al., 2019, Lindberg et al., 2018) of the LUCY model (Allen et al., 2011; Lindberg et al., 2013).


| | Daytime ($K_\downarrow > 5$ W m$^{-2}$) | | | | | | | Daily | | | | | | |
| | 2012 | | 2013 | | | | 2013: | 2012 | | 2013 | | | | 2013: |
| | Autumn | Winter | Spring | Summer | Autumn | Winter | Year | Autumn | Winter | Spring | Summer | Autumn | Winter | Year |
| | W m$^{-2}$ | W m$^{-2}$ | W m$^{-2}$ | W m$^{-2}$ | W m$^{-2}$ | W m$^{-2}$ | W m$^{-2}$ | MJ d$^{-1}$ | MJ d$^{-1}$ | MJ d$^{-1}$ | MJ d$^{-1}$ | MJ d$^{-1}$ | MJ d$^{-1}$ | MJ d$^{-1}$ |
|---|---|---|---|---|---|---|---|---|---|---|---|---|---|---|
| $K_\downarrow$ | 288.9 | 185.8 | 343.8 | 303.9 | 263.6 | 233.1 | 280.1 | 12.49 | 6.03 | 16.72 | 15.87 | 11.39 | 8.4 | 13.63 |
| $K_\uparrow$ | 44.8 | 48.2 | 49.3 | 43.4 | 39 | 42.3 | 41.5 | 1.94 | 1.57 | 2.4 | 2.27 | 1.69 | 1.53 | 2.02 |
| $L_\downarrow$ | 317.2 | 236 | 322.8 | 416.8 | 324.1 | 242.7 | 335.6 | 26.94 | 20.11 | 27.37 | 35.6 | 27.57 | 20.48 | 28.57 |
| $L_\uparrow$ | 396.1 | 295 | 404.1 | 469.6 | 396.4 | 318.3 | 404.6 | 32.87 | 24.62 | 33.5 | 39.6 | 32.95 | 26.02 | 33.8 |
| $Q_\downarrow$ | 606.1 | 421.8 | 666.6 | 720.7 | 587.7 | 475.8 | 615.7 | 39.42 | 26.14 | 44.09 | 51.46 | 38.96 | 28.88 | 42.19 |
| $Q^*$ | 165.2 | 78.6 | 213.1 | 207.6 | 152.4 | 115.2 | 169.6 | 4.61 | -0.04 | 8.19 | 9.6 | 4.33 | 1.33 | 6.37 |
| $Q_H$ | 47.1 | 44.6 | 85.5 | 33.8 | 44.8 | 53 | 52.6 | 1.75 | 1.13 | 3.85 | 1.59 | 1.65 | 1.45 | 2.29 |
| $Q_E$ | 56.4 | 14.5 | 48 | 108.5 | 45.7 | 11.5 | 58.9 | 2.7 | 0.65 | 2.57 | 6.09 | 2.22 | 0.52 | 3.13 |
| $Q_F$ | 23.5 | 32.8 | 23.4 | 24.7 | 23.6 | 32.1 | 25.3 | 1.57 | 2.09 | 1.59 | 1.72 | 1.58 | 2.01 | 1.72 |
| $\Delta Q_{S,res}$ | 95.6 | 55.5 | 106.3 | 108.5 | 83.8 | 84.3 | 92.8 | 2.07 | 0.17 | 3.48 | 4.55 | 1.89 | 1.45 | 3.08 |
| Albedo | 0.16 | 0.26 | 0.14 | 0.14 | 0.15 | 0.18 | 0.15 | 0.16 | 0.26 | 0.14 | 0.14 | 0.15 | 0.18 | 0.15 |
| $Q_H/Q_E$ | 0.83 | 3.08 | 1.78 | 0.31 | 0.98 | 4.6 | 0.89 | 0.65 | 1.72 | 1.5 | 0.26 | 0.74 | 2.81 | 0.73 |
| $Q_H/Q^*$ | 0.29 | 0.57 | 0.4 | 0.16 | 0.29 | 0.46 | 0.31 | 0.38 | -28.26 | 0.47 | 0.17 | 0.38 | 1.09 | 0.36 |
| $Q_E/Q^*$ | 0.34 | 0.18 | 0.23 | 0.52 | 0.3 | 0.1 | 0.35 | 0.59 | -16.42 | 0.31 | 0.64 | 0.51 | 1.08 | 0.49 |
| $\Delta Q_{S,res}/Q^*$ | 0.58 | 0.71 | 0.5 | 0.52 | 0.55 | 0.73 | 0.49 | 0.45 | -4.38 | 0.43 | 0.47 | 0.44 | 1.09 | 0.48 |
| $Q_H/Q_\downarrow$ | 0.08 | 0.11 | 0.13 | 0.05 | 0.08 | 0.11 | 0.09 | 0.04 | 0.04 | 0.09 | 0.03 | 0.04 | 0.05 | 0.05 |
| $Q_E/Q_\downarrow$ | 0.09 | 0.03 | 0.07 | 0.15 | 0.08 | 0.02 | 0.1 | 0.07 | 0.03 | 0.06 | 0.12 | 0.06 | 0.02 | 0.07 |
| $\Delta Q_{S,res}/Q_\downarrow$ | 0.16 | 0.13 | 0.16 | 0.15 | 0.14 | 0.18 | 0.15 | 0.05 | 0.01 | 0.08 | 0.09 | 0.05 | 0.05 | 0.07 |

### 3.5 Factors influencing energy balance fluxes

At MY, the impacts of precipitation on energy partitioning are obvious. As monthly rainfall from September to December in 2012 and 2013 are higher and lower than the normal (1991-2020) respectively, monthly daytime mean $\beta$ values of September-December in 2012 are correspondingly lower than those of the same period in 2013. Notably with no precipitation in November and December 2013, the $\beta$ in December 2013 reaches 7.40. This is 2.4 times greater than the December 2012 ($\beta= 3.08$). To our knowledge, this large $\beta$ value is one of the highest observed in suburban area (Fig. 12b), but also the central part of cities (Fig. 13), but still less than the values observed in central London ($\beta= 5$-$10$; Kotthaus and Grimmond, 2014) or Mexico City ($\beta= 8.8$; Oke et al., 1999).

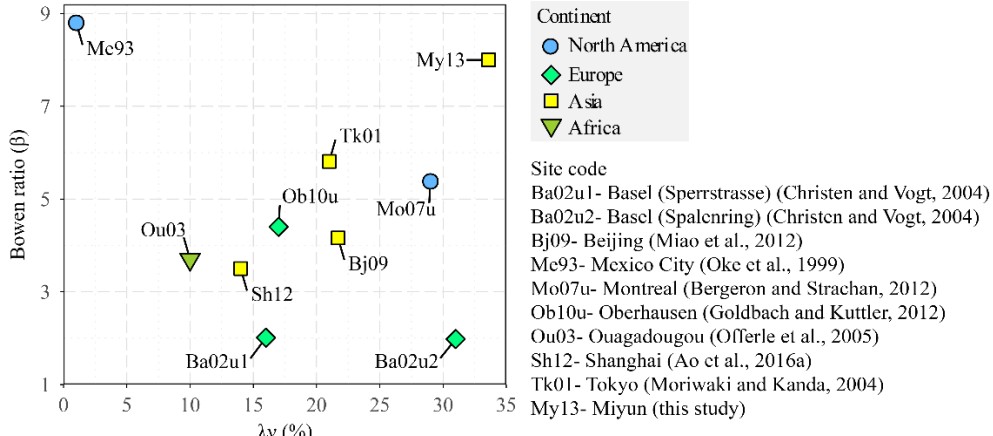

**Figure 13:** As Fig. 12 but winter daytime Bowen ratio for MY and for various urban sites with vegetation cover of 1-31%. At Bi09, Ba02u1, Ba02u2, Sh12, Bowen ratio is the mean value of Dec-Feb; At My13, Me93 and Mo07u, is the mean value of Dec; At Ob10u, is Jan; At Ou03 and TK01, is Feb. All Bowen ratios are accurately provided in the paper, except for Sh12 extracted from the plot and Mo07u extracted from ratios of $Q_H/(Q^* + Q_F)$ and $Q_E/(Q^* + Q_F)$.

Irrigation supplements precipitation and plays an important role in energy partitioning (Grimmond and Oke, 1986; Grimmond et al., 1996; Kokkonen et al., 2018). At MY, winter wheat and summer maize are the two predominant crops and cultivated in rotation. The growing season of winter wheat is from October to mid-June, while maize is planted in late June and harvested in the end of September every year. June and October are the intermittent months for the two crops. Irrigation is frequent during wheat growth in spring, because of the drought and little rain climatic conditions. As shown in Figure 14, the soil moisture of natural conditions and cropland were almost the same at the end of April (28th April). With only 3 days (26th, 27th and 28th May of 0.1, 0.4 and 0.2 mm, respectively) having rain in May

2013 (Fig. 2b), the soil moisture of natural conditions decreases dramatically in May, whereas of cropland increases in contrast. This indicates that cropland has irrigation to replenish water. For this reason, the May $\beta$ clearly presents a decreasing tendency even if there are more than 20 days of no rain (Fig. 15), as external water provided by cropland irrigation makes the available energy favor the latent heat flux. The monthly mean $\beta$ of 0.50 in May (Fig. 11), clearly indicates the importance of this additional source of water.

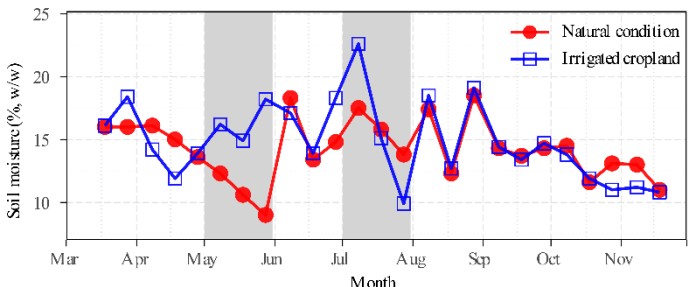

**Figure 14:** Gravimetric soil moisture (%) at a depth of 0.1 m measured on the 8th, 18th and 28th of each month at the MY site from 18 March to 18 November 2013 under natural condition and irrigated cropland. The shading is May and July 2013, respectively.

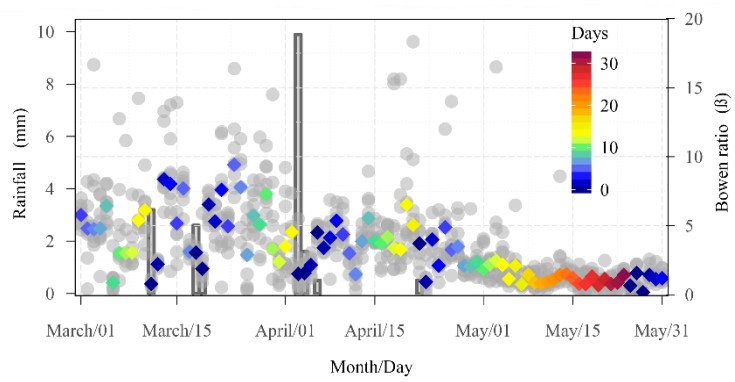

**Figure 15:** Daytime (circle) and median 30 min daytime (diamond) Bowen ratio by number of days since rainfall (colour) (right hand axis) and daily rainfall (bars, left-hand axis) for 1 March to 31 May 2013.

Crop growth and phenology are also important influences. Transpiration during crop growth releases a large amount of water into the atmosphere observed as $Q_E$ (e.g. Dou et al., 2019). The soil moisture in the irrigated cropland is higher than in the natural rainfed areas (Fig. 14). The gravimetric samples dates are critical relative to the timing of both irrigation and rainfall (e.g. 8 July 2013 cf. few

days later) as the soil moisture values can inverse, as cropland utilizes stored soil water via transpiration and evaporation.

In addition to these factors, land use/cover also plays a critical role in the energy budget. As the daily averaged source area is mainly covered by vegetation (60%) and impervious (87%) at 30-150° and 210-360° sectors of the tower respectively, and farmland and buildings are separately the most representative land use/cover types in the two sectors, observed data from these two sectors are selected to compare energy fluxes ratios and referred to as from farmland- and building-dominant directions

accordingly. At MY, normalizing by the incoming radiation ($Q_\downarrow = K_\downarrow + L_\downarrow$) the $Q_H/Q_\downarrow$ and $\beta$ ($Q_E/Q_\downarrow$) are always more (less) when the wind is from directions with more buildings (cf. cropland-dominated) irrespective of season (Fig. 16). Compared to impervious surfaces, the cropland surface obviously transforms between "dry" and "wet" more frequently and with longer transition time, due to cropland irrigation and water storage by soil. This results in a greater amplitude of $Q_H/Q_\downarrow$ and $Q_E/Q_\downarrow$ from the

cropland-dominated direction. The inter-quartile range and the differences between the mean and median of ratios are clearly greater from the cropland direction (Fig. 16). The temporal variations in $\beta$ after rainfall differ between these two direction types (Fig. 17). Water can drain relatively quickly after rain in the building-dominated direction because of the high proportion of impervious surface. Whereas, water can infiltrate into the soil (i.e. stored) so $\beta$ can remain near 1 for longer periods after rainfall (Fig. 17).


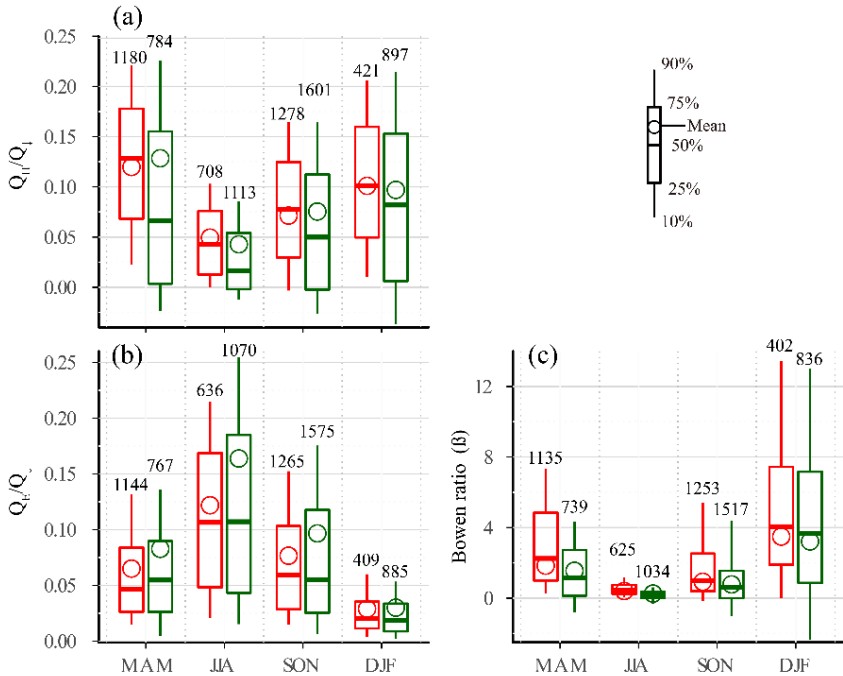

**Figure 16:** Median, IQR and mean flux ratio (30 min, number of periods indicated) by season when wind is from the building- (red) and farmland-dominant directions (dark green) (a) $Q_H/Q_\downarrow$ (b) $Q_E/Q_\downarrow$ and (c) $\beta$ (see Table 1 for definitions).

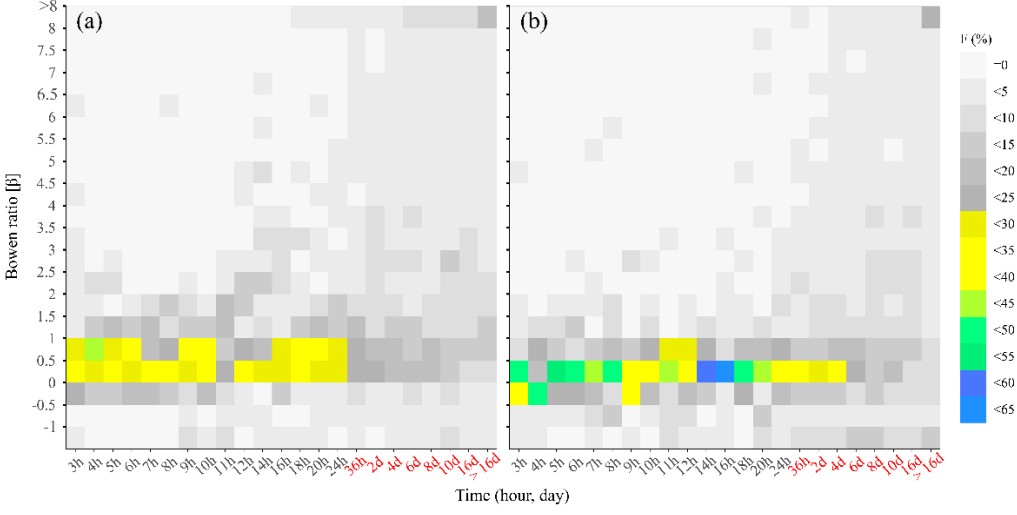

**Figure 17:** Frequency (F, %) of Bowen ratio $\beta$ values by time since last rainfall for (a) building-dominant (210-360°) and (b) farmland-dominant (30-150°) directions.

## 4 Conclusions

In this analysis of surface energy flux measurements for a suburb (MY) of Beijing over 16 months we gain a better understanding of surface–atmosphere dynamics.

All components of the radiation balance and net radiation ($K_\downarrow$, $K_\uparrow$, $L_\downarrow$, $L_\uparrow$, $Q^*$) have unimodal annual patterns, with higher values in summer (lower in winter) with the maximum monthly means in July (minima December / January).

      At MY, daytime sensible heat flux $Q_H$ is greatest in spring. However, it is smallest in summer rather than winter, unlike previous suburban studies. This is a because of the prevailing wind direction has extensive cropland, irrigation, and frequent rainfall in summer. All of these factors are thought to play a role. Nocturnal $Q_H$ is negative throughout the year and smaller in winter, so the minimum daily total $Q_H$ still occurs in winter.

      The latent heat flux $Q_E$ is positive throughout the day. The daytime maximum $Q_E$ is greatest in summer July (lowest in winter December) because of the influence of rainfall, irrigation, plant growth activity and available energy. Monthly median diurnal maxima of $Q_E$ vary from 10-248 W m$^{-2}$ during measurement period. These are close to both the smallest and largest values reported for suburban areas. The maximum monthly daily total $Q_E$ is in summer (July 2013) and minimum winter (December 2013).

      Daytime $Q_H/Q^*$ is lower in summer (higher in winter). Across the year it varies from 15-57%. $Q_E/Q^*$ has the opposite seasonal trends but similar annual range (6-56%). Summer daytime means of $Q_H/Q^*$ (0.16) are lower and $Q_E/Q^*$ (0.52) higher than reported in most suburban areas. While in winter daytime $Q_H/Q^*$ (0.41-0.57) is greater than reported for most suburban sites but $Q_E/Q^*$ (0.08 to 0.18) is similar to other suburban sites. Thus the storage heat flux $\Delta Q_S$ at MY is a small fraction of $Q^*$ because of the low building fraction. At the annual time scale, daytime $Q_H/Q^*$ (31%) and $Q_E/Q^*$ (35%) are very similar but for the whole day the proportions are 36% and 49%, respectively.

The large seasonal differences in precipitation and irrigation lead to a wide range in Bowen ratio ($\beta$) values across the year (0.26-7.40). Daytime mean $\beta$ in summer (0.32) are lower than winter (4.60); and are on the lower and higher end of the range in the literature for suburban sites. The annual mean daytime $\beta$ is 0.89 and the daily value is 0.73; thus $Q_E$ dominates available energy.

      The results confirm the combined importance of precipitation, irrigation, crop/vegetation growth and land cover as being key factors affecting energy partitioning in MY. These findings will help to enhance the understanding of the surface-atmosphere energy exchange over Chinese suburban areas,

provide observational data for model verification and parameterization scheme improvement, and be a reference for formulating policies to mitigate the adverse effects of urban climate and climate change.

**Appendix A: Anthropogenic heat flux estimation**

LQF provides a method to calculate anthropogenic heat flux $Q_F$ (Gabey et al., 2019) based on population, vehicle, energy consumption and air temperature data. It estimates heat released from buildings $Q_{F,B}$, traffic $Q_{F,V}$, and human metabolism $Q_{F,M}$ (Allen et al., 2011; Lindberg et al. 2013). In this study, we determine appropriate temperature response coefficient for the MY area.

**A1 Temperature response coefficient**

In LQF, the mathematical expression of $Q_{F,B}$ is as follows:

$$Q_{F,B} = \rho_{pop} f_b E_{B,b} \tag{A1}$$

where $\rho_{pop}$ is the population density (capita ha$^{-1}$) and $E_{B,b}$ is daily building energy consumption (kWh day$^{-1}$ per capita). The latter, from daily electricity consumption data, accounts for 16% of the total energy consumption in MY (Table A1).

The coefficient, $f_b$, captures the air temperature response to of energy consumption. Given large differences in energy consumption between countries and between cities (e.g. Lindberg et al. 2013), it is preferable to determine the appropriate local $f_b$ as it plays a critical role in building anthropogenic heat emissions. Here $f_b$ varies with air temperature $T_a$:

$$f_b = C + A_c(T_a - T_b), \quad T_b < T_a < T_{max}$$
$$f_b = C, \quad T_a = T_b$$
$$f_b = C + A_h(T_b - T_a), \quad T_{min} < T_a < T_b \tag{A2}$$

Several key parameters to determine $f_b$ are shown in Fig. A1. Here $T_b$ is the air temperature when the energy consumption is the lowest. If the air temperature is higher (lower) than $T_b$, more energy will be consumed due to cooling (heating). $A_c$ ($A_h$) is the building energy consumption thermal response 570 slope for cooling when air temperature above (below) $T_b$, also known as a cooling (heating) coefficient. $C$ is the minimum energy consumption. $T_{max}$ and $T_{min}$ are threshold values for air temperature. When

air temperature is beyond this range, energy consumption has reached saturation and no longer increases with air temperature changes.

A new set of temperature response parameters applicable to this region is obtained by using daily
air temperature and electricity consumption data for the Miyun district in 2012-2013 (Table A1). The resulting parameters are given in Table A2. The energy consumption response to temperature changes is a "V"-shape curve in Miyun, rather than a "U"-shape as seen in Shanghai city (Ao et al., 2018). The cooling coefficient $A_c$ in MY is lower than that in Shanghai city (0.04 ℃$^{-1}$), which is attributed to relatively short period air conditioning in MY given the cooler summer nights caused by topography and
continental monsoon climate. However, in winter with colder air temperatures and a longer heating period the heating coefficient $A_h$ for MY is higher than for Shanghai ($A_h$ =0.01℃$^{-1}$) (Ao et al., 2018).

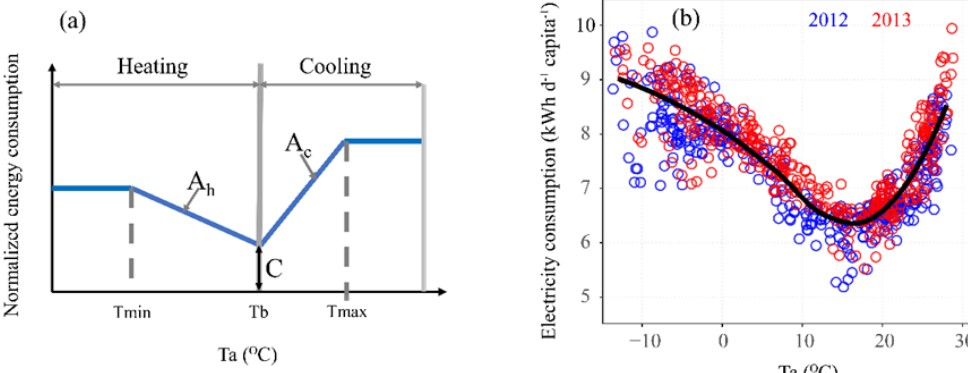

**Figure A1:** Energy consumption response to air temperature: (a) general response function (see text for definitions) and (b) data for Miyun District (Miyun District daily electricity consumption; kWh day$^{-1}$ per capita; State Grid
Beijing Electric Power Company, http://www.bj.sgcc.com.cn/) normalized by population (245 people km$^{-2}$ in 2012 and 246 people km$^{-2}$ in 2013; Beijing Miyun Statistical Yearbook, 2013; 2014) and MY daily mean air temperature (Ta) for 2012-2013 with general trend (black, loess curve).

**Table A1:** Energy consumption in Miyun (Beijing Miyun Statistical Yearbook, 2013; 2014) in ton coal equivalent (TCE) is converted to kilowatt hours assuming 1 TCE = 8141 kWh (Kyle's Converter, 2017).

| Year | Energy consumption (10$^4$ TCE) | Energy consumption (100 million kWh) | Electricity consumption (100 million kWh) | Electricity consumption / Energy consumption |
|------|------|------|------|------|
| 2012 | 105.0 | 85.48 | 13.62 | 0.1593 |
| 2013 | 109.3 | 88.98 | 14.34 | 0.1612 |

**Table A2.** Parameters of energy consumption response to air temperature (equation A.2)

| Site | $T_b$ (°C) | $A_h$ (°C$^{-1}$) | $A_c$ (°C$^{-1}$) | C - | $T_{max}$ (°C) | $T_{min}$ (°C) |
|------|------|------|------|------|------|------|
| MY | 16 | 0.023 | 0.030 | 0.83 | 50 | -20 |

**A2 Vehicle numbers**

In LQF, $Q_{F,V}$ is calculated as a function of vehicle numbers, traffic speed, and time (Ao et al., 2018). The average Beijing vehicle numbers per 1000 capita in 2012 and 2013 (245 people $km^{-2}$ in 2012 and 246 people $km^{-2}$ in 2013; Beijing Miyun Statistical Yearbook, 2013; 2014) are: cars = 196.9 and 201.7, motorcycles = 11.7 and 11.7, and freight vehicles = 42.6 and 43.7. For all vehicles it is assumed: an average vehicle speed = 48 km $h^{-1}$ and a fuel of petrol to be used. Diurnal variation of vehicle numbers for weekdays, weekends, and holidays for Shanghai (Ao et al., 2018) are used as no such data are available for Miyun or Beijing. This will cause additional uncertainties. Given the $Q_{F,V}$ values are small (Section 3.2 and Figure S2b), these additional uncertainties could be ignored.

**A3 Population density**

The Miyun Meteorological Station is located within the "Gulou Street" administrative division of Miyun District. The population density of the "Gulou Street" area was 5657 and 5702 people $km^{-2}$ in 2012 and 2013, respectively (Beijing Miyun Statistical Yearbook, 2013; 2014). This population density is much greater than in the Gridded Population of the World, version 4 (GPWv4; CIESIN, 2016) of about 1007 people $km^{-2}$ around the study site. This is why Lindeberg et al. (2013) and Gabey et al. (2019) indicate the importance of updating data locally where ever possible.

**Appendix B: Storage heat flux estimation using OHM**

One of the two methods (Sect. 2.5) used in this study to determine the storage heat flux is the objective hysteresis model (OHM), which uses coefficients by land cover type ($a_{1,i} - a_{3,i}$) with net all-wave radiation $Q^*$ (Grimmond et al., 1991; Meyn and Oke, 2009):

$$\Delta Q_{s,ohm} = \sum_{i=1}^{n}\{a_{1i}Q^* + a_{2i}(\partial Q^*/\partial t) + a_{3i}\}f_i \tag{B1}$$

where $t$ is time and, $f_i$ is the area fraction covered by $i$th land cover type. Given the differences in land cover around MY flux tower, the plan area fractions by 30° wind sector footprint area are used to calculate $\Delta Q_{s,ohm}$ by direction (Fig. B1).

The coefficients used (Table B1) include values obtained from 30 min measured data of net radiation $Q^*$ and soil heat flux $Q_G$ in Weishan Farmland Experimental Station (116° 09′ E, 36° 39′ N). Weishan

and MY have similar climate background, the same wheat/maize rotation pattern, and consistent growth cycle (Lei and Yang, 2010; Lei et al., 2018).

The Weishan station has a 10 m tower with a four-component radiometer (CNR1, Kipp & Zonen, Netherlands) installed at 3.5 m agl and two soil heat flux plates (HFP01SC, Hukseflux, Netherlands) installed at a depth of -0.03 m. The instrument data are recorded at 10-min intervals using a CR10X data logger (Campbell Scientific, USA). The soil heat flux $Q_G$ data are the average of two measurement sites on the east and west of the tower.

For application convenience, the cropland coefficients are determined by season. A random, but arbitrary 70% of data are selected in each season to fit the coefficients. The remaining 30% is used to evaluate the simulation results. The results are good, but poorest in winter (Fig. B2). These values are used for the farmland (Table B1). Some large observed $Q_G$ are not reproduced by the OHM model, as they are the maximum values and appear simultaneously with the peak of $Q^*$ on that day. Generally, the maximum $Q_G$ appears later than that of $Q^*$, for the fitted $a_2$ of farmland being negative in all seasons

(Table B1). This phenomenon has been reported in previous studies, i.e., OHM estimates perform satisfactorily in the mean but miss short-term variability (Grimmond and Oke, 1999; Roberts et al., 2006).

Building coefficients also vary with season (Table B1). Here we use data from Nanjing city (Wang et al., 2008), with an average of summer and winter values used in spring and autumn. Other coefficients are used year round. For the Road/Impervious areas an average of literature values is used (Table B1).

The MY site coefficients vary with season and time of day (Fig. B3). The $a_1$ and $a_2$ are larger in the afternoon, as the wind comes from the southwest direction where artificial underlying surfaces dominate (Fig. 2g-j; Fig. B1). The $a_1$ and $a_2$ of various impervious surfaces are relatively high (Table B1). The winter $a_1$ and $a_2$ are smaller than other seasons throughout the day. This is because the prevailing wind in winter comes from the farmland-dominant direction, while the $a_1$ and $a_2$ of farmland

in winter are the smallest (Table B1). The $a_3$ is exactly the opposite of $a_1$ and $a_2$.

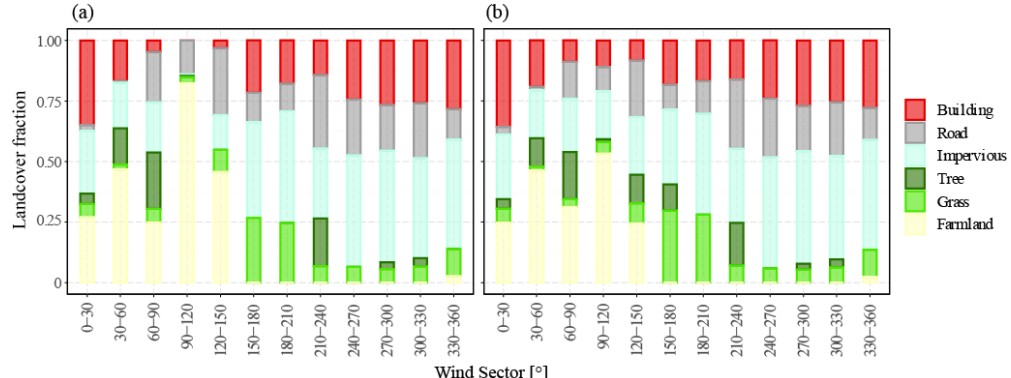

**Figure B1:** Land cover fraction for 30º wind sectors, but varying footprint length (Section 2.4), around the MY flux tower for (a) day ($K_\downarrow > 5$ W m$^{-2}$) and (b) night.

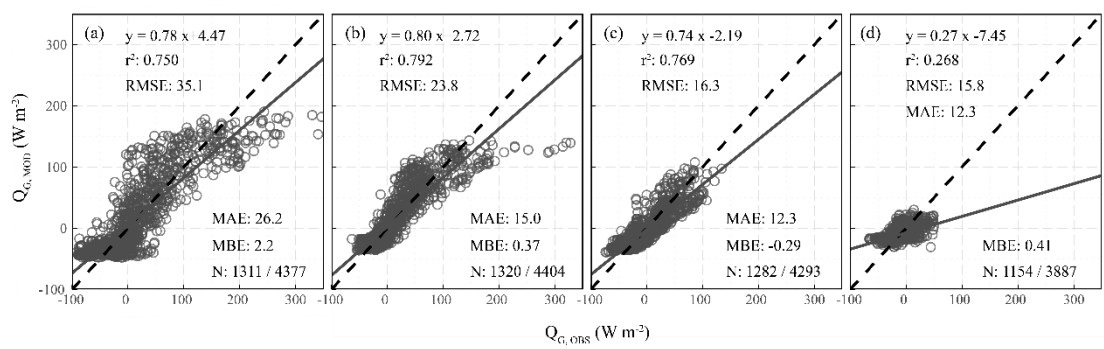

**Figure B2:** Weishan station modelled and observed soil heat flux, with statistical evaluation metrics (root mean square error (RMSE), coefficient of determination ($r^2$), mean absolute error (MAE) and mean bias error (MBE). N: evaluation data / total observations available) for (a) spring, (b) summer, (c) autumn and (d) winter.

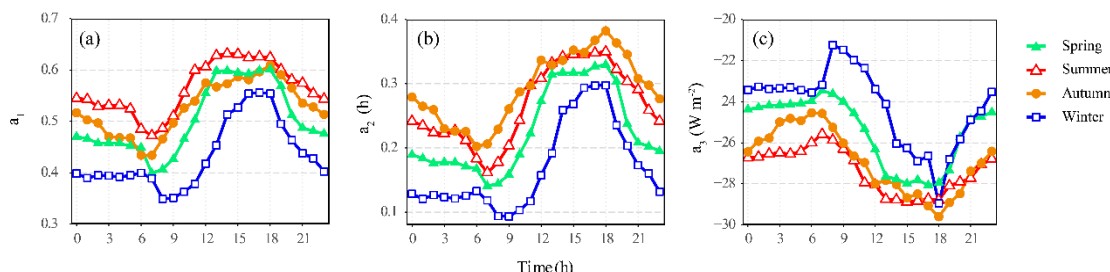

**Figure B3:** Seasonal diurnal variation of OHM coefficients (a) $a_1$, (b) $a_2$ and (c) $a_3$ for Miyun.

**Table B1:** Coefficients used for the OHM (eqn B1) are from literature or derived in this study (*).

| Surface cover | | Coefficients | | | Source |
|---|---|---|---|---|---|
| | | $a_1$ | $a_2$ (h) | $a_3$ ($W\ m^{-2}$) | |
| **Vegetation** | *Tree* | 0.11 | 0.11 | -12.3 | McCaughey (1985): Mixed forest |
| | *Grass* | 0.32 | 0.54 | -27.4 | Doll et al. (1985): Short grass |
| | Wheat *Spring* | 0.29 | -0.19 | -20.7 | Weishan station, * |
| | Wheat/Maize *Summer* | 0.23 | -0.07 | -17.2 | Weishan station, *: Wheat: June; Maize: July and August |
| | Maize/Wheat *Autumn* | 0.21 | -0.14 | -17.6 | Weishan station, *: Maize: September; Wheat: October and November |
| | Wheat *Winter* | 0.07 | -0.29 | -11.8 | Weishan station, * |

| Building | Summer | 0.83 | 0.52 | -16.95 | Wang et al (2008): Concrete roof |
|---|---|---|---|---|---|
| | Winter | 0.86 | 0.27 | -9.64 | Wang et al (2008): Concrete roof |
| | Spring/ Autumn | 0.845 | 0.395 | -13.295 | Summer and winter mean |
| Road/Impervious | | 0.36 | 0.23 | -19.3 | Narita et al (1984): Asphalt |
| | | 0.81 | 0.48 | -79.9 | Doll et al (1985): Concrete |
| | | 0.85 | 0.32 | -28.5 | Asaeada and Ca (1993): Concrete |
| | | 0.64 | 0.32 | -43.6 | Asaeada and Ca (1993): Asphalt |
| | | 0.82 | 0.68 | -20.1 | Anandakumar (1999): Asphalt |
| | | 0.696 | 0.406 | -38.28 | Average of road/impervious, * |

**Data availability**

The observation data used in the study are available from the corresponding author with permission (E-
mail: jxdou@ium.cn). We downloaded flux data of Yucheng agricultural experiment station from National Ecosystem Science Data Center, National Science & Technology Infrastructure of China website: http://www.nesdc.org.cn (last access: 3 August 2023;

https://doi.org/10.12199/nesdc.ecodb.chinaflux2003-2010.2021.yca.005, National Ecosystem Science Data Center, 2021).

**Author contribution**

JD conducted the eddy covariance measurement, carried out the data analyses, and wrote the draft. SG supervised the scientific interpretation of the results and polished the writing. SM, BH, HL and ML provided key data sets and contributed to the anthropogenic heat flux estimation, and OHM coefficients fitting.

**Competing interests**

The authors declare that they have no conflicts of interest.

**Acknowledgments**

This research was supported by the Youth Beijing Scholars Program (Grant No. 2018-007), the National Natural Science Foundation of China (Grant No. 41505102), Key innovation team of
China Meteorological Administration (No. CMA2022ZD09), the Beijing Natural Science Foundation (Grant No. 8212026) and China Scholarship Council (CSC No. 201805330002). We offer special thanks to Weishan experiment station for providing observation data of net radiation and soil heat

flux in farmland used to fit OHM parameters in the paper. We thank all those who have supported the observation analyzed in this project. SG thanks the Newton Fund/Met Office CSSP-China project and

ERC urbisphere (855005).

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
