# Peer review of "Surface energy balance fluxes in a suburban area of Beijing: energy partitioning variability"

_Atmospheric Chemistry and Physics, 2021_

## Author Comment (AC2)

We are very grateful for the comments and suggestions provided by the reviewers. These comments and suggestions are very constructive and helpful. The paper has been updated in the light of these.

In the section below, reviewers' comments are in black, our responses in blue and new text in red. Figure numbers/Lines etc. referred to the revised manuscript unless otherwise indicated.

**Comments of Reviewer #1**

**General comments**

This manuscript describes 16 months of radiation and surface energy balance fluxes observed at a mixed land-use site near Beijing (Miyun). The highly complex site is classified as suburban, consisting of moderately dense residential buildings (6 storeys) to the west of the tower and a mixture of farmland, roads and buildings to the east. The observed radiative, sensible heat and latent heat fluxes are supplemented with estimations of the remaining energy balance terms using simple models. The methodology and tools used are similar to other urban climate studies, but the characteristics of the Miyun site are rather atypical which makes this a potentially interesting addition to the urban literature.

Analysis and interpretation of this complex site is challenging. To some extent the methodology takes these challenges into account (e.g. via source area analysis, consideration of storage fluxes for farmland as well as the built environment, estimation of anthropogenic heat emissions, measurements of soil moisture). However, the analysis is quite brief which limits the impact and usefulness of the study. I believe this is partly due to the broad approach used and missing further investigations which would add weight to the findings.

One of the main conclusions of this work seems to be the range of factors affecting the surface energy fluxes, yet the bulk of the paper only really assesses monthly and average monthly diurnal variations with the effects of different meteorological conditions, surface cover fractions, anthropogenic influences, and spatial and temporal variability all combined. In addition, these various factors don't seem to be dealt with consistently. For example, the prevailing wind is from the east (so the source area is predominantly farmland, small residential buildings and roads) but large anthropogenic heat fluxes are calculated for the study site as a whole. Also, despite the variation in building and tree heights around the site it seems a single value has been used for the roughness length and displacement height for all wind directions. The storage heat flux estimation attempts to account for the different source area characteristics but most of the analysis (except Fig 11 and 12 right at the end) is very broad (i.e. monthly differences without any stratification by wind direction). In my opinion, the manuscript needs to deal with these challenges in a more consistent way and refine the analysis to provide insight into these physical controls.

Substantial revisions are needed in order to develop this manuscript into a valuable contribution. The language is mostly understandable but often difficult to follow and there are many mistakes. Therefore the revised manuscript will need English language editing (both for small errors and to

improve readability and structure). The main messages of the paper should be given more attention, in particular the comparisons with previous suburban studies. This is one of the purposes of this study but in many places the analysis is very brief and offers little new insight (see below for details). At the same time, unnecessary and unimportant details should be removed (e.g. overly detailed descriptions of building heights; Table 1) or conveyed more efficiently (e.g. with a figure, table or single sentence), or, if important, these details would be better consolidated with the main text (e.g. some of the supplementary information).

I have tried to make suggestions for where particular changes are needed. These range from major topics to more minor issues. I have also provided some language corrections for the first half of the paper but stopped after Section 3.1 as there are too many to expect a reviewer to detail them all.

We are very grateful for the particularly detailed and excellent comments from the reviewer. These comments and suggestions are very enlightening and helpful. Please see our point-by-point responses below.

**Specific comments - major**

**Single values used for roughness length and displacement height?**

L158-72: There is a lot of text given over to the description of this complex site – the variation in land cover, building height and vegetation height. Since there is considerable variation in building and vegetation height spatially (around the tower) and temporally (with season), can it really be justified to use single value for the roughness length and displacement height? Or are these single values only used as an initial estimate for the footprint model? Although the initial estimates for the iterative footprint calculation may not be crucial, the roughness length and displacement height used for the flux processing and to calculate the stability parameter could impact the results. It is not clear what has been done here.

Only single $z_0$ (2.9 m) and $z_d$ (4.1m) are used to calculate the turbulent fluxes source areas, the stability parameter ($\zeta$), and the flux data processing in this study. $z_0$ and $z_d$ do not affect the turbulent heat fluxes ($Q_H$ and $Q_E$) but do impact the calculated the turbulent fluxes source areas and the stability parameter ($\zeta$).

We compared $Q_H$, $Q_E$ and $\zeta$ values by using single $z_0$ (2.9 m) and $z_d$ (4.1 m) with those by using the various $z_0$ and $z_d$ values, to prove that $z_0$ and $z_d$ values do not impact study results shown in the submitted manuscript.

The Macdonald et al. (1998) $z_0$ and $z_d$ estimated for 20° wind sectors (radius = 597 m) vary between 0 and 8.1 m ($z_0$), and from 0 to 13.3 m ($z_d$) (Figure rev.1). The radius is chosen based on the 2013 median 90% flux source area extent derived using a $z_0$ (2.9 m) and $z_d$ (4.1 m).

[Figure]

Figure rev1. The roughness length for momentum $z_0$ (a) and zero plane displacement height $z_d$ (b) for 20º wind sectors within the 597 m radius around MY flux tower by using the Macdonald et al. (1998) method.

The data in May 2013 were selected to compare the processing results of applying various and single $z_0$ and $z_d$ values. It was found that the $Q_H$ and $Q_E$ obtained by the two methods were highly consistent (See the fitting results at the end of this paragraph), indicating that different $z_0$ and $z_d$ values do not affect $Q_H$ and $Q_E$. Although ζ values are influenced by $z_0$ and $z_d$, they would not change the judgment of the variation of turbulence being stable or unstable. Given that it has good consistency in land use/cover types in the same direction around the Miyun site, small differences in the flux source areas (difference in radius < 15 m between using various and single $z_0$ and $z_d$) should have less impact on the land cover fraction. Therefore, fluxes data displayed in the submitted manuscript were reliable.

Various vs Single $z_0$ and $z_d$ fitting results:

$Q_H$:   y = 0.999*x-0.01    $r^2$= 0.9999912
$Q_E$:   y = 1.0*x-0.02     $r^2$ = 0.9999932
ζ :   y = 1.07*x+0.02    $r^2$ = 0.997

Macdonald R, Griffiths R, Hall D (1998) An improved method for the estimation of surface roughness of obstacle arrays. Atmos Environ 32:1857-1864.

**Further analysis of radiation would be useful**

L209-249: This part could be made more concise as there are not very many new or interesting results here that go beyond what is widely known. L237-239 should be deleted if it is irrelevant or rephrased so that the reason for including this statement is clear to the reader. The period with very little incoming shortwave radiation in early June 2013 is potentially interesting and could be investigated further (e.g. with a zoomed-in timeseries and further discussion). This would also help to show that the observed data realistically capture the conditions on these days and avoid any uncertainty in the mind of the reader that there could have been an instrument issue at this time.

Line 237-239 (Line 271-273 in the revised manuscript) has been deleted.

Timeseries plot (Figure 4 in the revised version) of radiation components, rainfall, air temperature, specific humidity, and soil temperature for the very little incoming shortwave radiation period 3-10 June 2013, with 4 June selected as the case for its daily rainfall being the largest during this period, has been added in the revised manuscript. Accordingly, the statements that the incoming shortwave

radiation decreased sharply or even fell to 0 due to rainfall during this period has been added to in revised manuscript (Figure 4 c, d, j, k).

Text added (Line 240-242):

With rain everyday from 4 to 9 June, the $K_\downarrow$ maxima was < 220 W m$^{-2}$ (Fig. 4c-d). The $K_\downarrow$ was 0 W m$^{-2}$ between 11:00-12:00 and 16:00 on 4 June during rainfall (Fig. 4j-k).

[Figure]

**Figure 4:** Time series of (a, h) air temperature at 2 m and 36 m and surface temperature, (b, i) specific humidity at 2 m and 36 m, (c, j) rainfall, (d, k) incoming shortwave radiation, (e, l) outgoing shortwave radiation, (f, m) incoming and outgoing longwave radiation and (g, n) net radiation for the period (a-g) 3-10 June 2013 and (h-n) June 4, 2013 at Miyun. Rainfall is 1 hour and other data are 30 min resolution.

In L247-9 December is highlighted as having the lowest net radiation because of the snow cover, but other winter months have similarly low values and snow was also present for the January data.

The text should be reconsidered to provide a more balanced and accurate assessment of what the data shows (i.e. does snow cover really play as important a role as is implied?).

Five snowfalls occurred in December 2012, reducing $K_\downarrow$, resulting in monthly mean $Q^*$ being $< 0$ W m$^{-2}$ in December. As the 28 December 2012 snowfall stayed on the ground until 8 January 2013, the January $K_\uparrow$ values are impacted despite not new snow occurring.

The monthly mean $K^*$, $K_\downarrow$ and $K_\uparrow$ in December 2012 (19.2 W m$^{-2}$, 24.6 W m$^{-2}$, 5.4 W m$^{-2}$ respectively) are less than those in January. Whilst the December $L_\downarrow$ and $L_\uparrow$ are larger than in January, with $L^*$ 11.7 W m$^{-2}$ higher (cf. January), it fails to offset $K^*$ resulting in the December $Q^*$ being -0.5 W m$^{-2}$, or 7.6 W m$^{-2}$ less than January.

[Figure]

Figure rev 2. The monthly mean diurnal course of all radiation components measured for December 2012 (left panel) and January 2013 (right panel). Numbers are daily averages in W m$^{-2}$. The total of the blue shaded area is the net shortwave gain $K^*$, the total of the light salmon hatched area is the net longwave loss $L^*$.

Text "(i.e. increasing $K_\uparrow$)" changed to "(i.e. decreasing $K_\downarrow$)" in the line 283 of the revised version.

Figure S1 could be moved to the main text and analysed in more detail, for example is this asymmetrical diurnal pattern in albedo also seen for days without snow cover? What is the physical explanation? Could this (and the need to set the shortwave components to zero at night) indicate a potential levelling issue with the radiometer?

Figure S1 in the submitted manuscript becomes Figure 5 in the revised version.

A new Figure S1 has been added in the revised supporting information. There is an asymmetrical diurnal pattern in albedo, which becomes more obvious after September once the solar elevation decreases (Figure S1c) and most pronounced in December, when no precipitation occurs, and the solar altitude angle is minimal. We believe surface heterogeneity explains this asymmetry rather than radiometer levelling. As can be seen from Figure S1a, there is a basketball court in the southeast of the radiometer, and its reflection of sunlight should be different from that of other ground surfaces. In addition, the specular reflection of the window glass of the office building about 9 m away to the

north of the tower may also be the reason for the asymmetry of the albedo (Figure S1a-b). The building is mainly in the north-by-east direction of the tower, and the building body inclines slightly to the north, which makes the radiometer more vulnerable to receiving the glass-reflected light most of the daytime, especially before noon. When the solar altitude changes in a certain range, the radiometer can receive the reflected light from the glass window, resulting in greater albedos during a certain period of daytime and asymmetry of the albedo.

Text added (Line 247-253):

When there is no snow on the ground an asymmetry of albedo still exists. With smaller solar elevation, the asymmetry is more pronounced (Fig. S1). Surface heterogeneity is thought to explain this as a basketball court is in the field of view of the radiometer (southeast, Fig. S1a) and albedo will differ from impervious and vegetation surfaces in other parts of the FOV (Fig. S1a), including specular reflection from glass. The building with windows is ~9 m north of the tower, increasing the albedo before noon (cf. afternoon) (Fig. S1b-c).

[Figure]

Figure S1. Study site (a) aerial view around MY flux tower (red circle, tower location) (Google Earth 2022); (b) photo of flux tower and the building to the north of the tower; (c) median diurnal variations (points) and interquartile ranges (shading) of albedo for June, September, October, November, and December 2013.

**Further analysis of anthropogenic heat flux needed**

L259-263: Little evidence is provided in support of these QF values being judged reasonable and inclusion of these data is of little value without further analysis. What could be the reason for differences compared to the other suburban studies suggested here? How does the Miyun site compare to the Basel, Montreal and Swindon sites in terms of temperature (heating/cooling degree days), population density, use of air conditioning. Why not add a few sentences about the relative contributions to QF (i.e. what proportion is from buildings (for heating and for cooling), traffic and human metabolism)? How do these contributions compare to values reported for other sub/urban sites? In Appendix A it seems to be suggested that the traffic contribution may be overestimated (L530-1) – how does this fit with L263?

Anthropogenic heat flux ($Q_F$) analysis (Section 3.2) is revised (Line 309-331).

- As the annual Basel $Q_F$, determined by energy balance residual ($Q^* + Q_H + Q_E$) does not

have temporal variations, we remove it from the discus (Line 327-328).

- Text (Line 298-306) bout the heat released from buildings, traffic and human metabolism and their contributions to the $Q_F$ added.

Text added (Line 309-317):

The median $Q_F$ vary between 5 and 39 W m$^{-2}$ diurnally across all of the months (Fig. 6). Heat released from buildings ($Q_{F,B}$) dominated $Q_F$, with its diurnal median value ranging from 4.6 to 36.4 W m$^{-2}$ and accounting for 85-95% of $Q_F$ (Fig. S2a). Contribution of vehicle emission ($Q_{F,V}$) is small (1-9% of $Q_F$). The maximum $Q_{F,V}$ value during the morning rush hour (8:00-9:00) is only 1.5 W m$^{-2}$, since the house is usually close to the working place or school and residents do not rely on cars for daily travel or commuting at MY (Fig. S2b). The diurnal variation of human metabolism ($Q_{F,M}$) is the constant within a year because of the fixed population density (5657 people km$^{-2}$ in 2012 and 5702 people km$^{-2}$ in 2013). Like other studies, $Q_{F,M}$ is small, with values ranging between 0.4-1 W m$^{-2}$ and contributions being 5-8%.

- Comparison between sites, and explanations for differences added (Line 318-325).

Text added (Line 318-325):

The fluxes are larger in summer and winter associated with cooling and heating needs, respectively. Our $Q_{F,B}$ values are larger than estimated in other suburban sites, such as Montreal during winter (Bergeron and Strachan, 2012) and Swindon (Ward et al., 2013), even the air temperature in Montreal is lower in winter. The reason is that Miyun has a greater mean building height (13.1 m) and building cover (21%). Moreover, buildings include hospital and office buildings, which usually have greater energy consumption and emissions than residential buildings. However, our $Q_{F,V}$ values are lower than those reported in Montreal and Swindon, due to the small number of motor vehicles and various travel modes at Miyun.

- Diurnal variation in vehicle numbers, used in calculating heat released from traffic ($Q_{F,V}$) are from Shanghai (Ao et al., 2018). However, with no obvious morning rush-hour in Miyun we expect $Q_{F,V}$ to be overestimated but as the daily maximum $Q_{F,V}$ value is only 1.5 W m$^{-2}$ during the morning rush hour (Section 3.2 and Figure S2b), the following sentence has been added to the revised manuscript (Line 679-680, Appendix A).

Text added (Line 679-680):

Given the $Q_{F,V}$ values are small, these additional uncertainties could be ignored.

[Figure]

Figure S2. Monthly median diurnal pattern (points) with inter-quartile ranges (shading) of (a) anthropogenic heat flux ($Q_F$) and heat released from buildings ($Q_{F,B}$), (b) heat released from traffic ($Q_{F,V}$) and human metabolism ($Q_{F,M}$).

**Storage heat flux methodology and analysis**

It is good to see that a lot of consideration has been given to the estimation of this important term in the urban surface energy balance and the discussion includes potentially useful results. However, this section needs to be better structured so that the key findings come across. The language needs to be improved (in terms of the grammar of individual sentences as well as the overall story) and it the manuscript would be much more readable if the important results from the supplementary information were included in the main text.

The methodology (i.e. various approaches used to calculate QS_OHM) should be more clearly set out to guide the reader – at the moment the text jumps between the impact of seasonality, time of day, source area and methodology from sentence to sentence. More explanation of Fig S2 is needed so that the reader immediately realises that the monthly and diurnal variation seen is due to the combination of (i) different seasonal coefficients and (ii) different source area composition. A couple of sentences explaining the results would also help (e.g. 'a1 is larger in the afternoons because…').

We made significant modifications to the calculation method and analysis of heat storage flux. Please refer to the response to reviewer's comments and the revised manuscript below for specific modifications.

The original Figure S2 has been moved to Appendix B and ordered as Figure B3. The interpretation of the OHM model and the explanation of Fig B3 has been added in Appendix B (Line 716-721).

Text added (Line 716-721):
The MY site coefficients vary with season and time of day (Fig. B3). The $a_1$ and $a_2$ are larger in the afternoon, as the wind comes from the southwest direction where artificial underlying surfaces dominate (Fig. 2g-j; Fig. B1). The $a_1$ and $a_2$ of various impervious surfaces are relatively high

(Table B1). The winter $a_1$ and $a_2$ are smaller than other seasons throughout the day. This is because the prevailing wind in winter comes from the farmland-dominant direction, while the $a_1$ and $a_2$ of farmland in winter are the smallest (Table B1). The $a_3$ is exactly the opposite of $a_1$ and $a_2$.

L275-7: Given the expectation that QS_RES is an upper limit for the storage heat flux and the well-known issue of energy balance under-closure, is the neglected biomass heat storage flux the main explanation here? The text implies that this is the message the reader should take away, but from the results shown and brief analysis I am not yet convinced that this is an important term or one of the main reasons for the differences. This paragraph needs to better introduce some of the other considerations that follow.

   This paragraph has been revised. We have amended the statement that the uncounted biomass heat storage is the main explanation for the underestimated $\Delta Q_{S,ohm}$. Of course, the neglected biomass heat storage is also one of the reasons for the underestimation of $\Delta Q_{S,ohm}$, but its impact is small.
   We believe that the main reasons why $\Delta Q_{S,ohm}$ is underestimated in autumn and winter (Nov-Feb) include the following:
   1. The $\Delta Q_{S,ohm}$ of farmland is underestimated in autumn and winter.
   The easterly wind prevails in autumn and winter, and the farmland covers a greater proportion of the flux source area. Hence the $\Delta Q_{S,ohm}$ of farmland have a great impact on the overall $\Delta Q_{S,ohm}$ estimation within the footprint area. The OHM parameters of farmland are fitted based on the observation data of net radiation $Q^*$ and soil heat flux $Q_G$. Given that $Q_G$ is only a part of the heat storage flux $Q_S$, the modeled $\Delta Q_{S,ohm}$ of farmland will be lower than the true $Q_S$, even if the fitted effect of OHM parameters is ideal. However, the modeled $Q_G$ of the farmland by the OHM is less than the observed $Q_G$, especially in winter. So, the $\Delta Q_{S,ohm}$ is underestimated in autumn and winter.
   2. OHM parameters may not be applicable to autumn and winter.
   Except for farmland and roof, OHM coefficients of other land cover types used in this study are fitted from the spring or summer observation dataset, while these coefficients may not be applicable to autumn and winter and cause the $\Delta Q_{S,ohm}$ to be underestimated in autumn and winter.
   In addition to the reasons mentioned above, mismatching the source area between radiation and turbulent fluxes will cause the uncertainty of $\Delta Q_{S,ohm}$ estimation, which may also lead to $\Delta Q_{S,ohm}$ underestimation in autumn and winter.

L278-301: It is interesting to see the consideration of other possible factors influencing the storage heat flux estimates such as energy balance under-closure and uncertainties in QF. Differences in footprint between radiometer and flux tower as well as the performance of the OHM model shown in Fig B2 (especially concerning the large values of observed soil heat flux that are not reproduced by the model) should also be discussed here.

   The discussion on uncertainties in storage heat flux estimation caused by footprint mismatch between radiation and turbulent fluxes is added in the revised version (L407-408); Meanwhile, a new plot, showing the monthly median diurnal variations of $\Delta Q_{S,ohm}$ within the footprint source

area of radiation and turbulent flux respectively, has been added to the supplement materials (Figure S4) to explain this point.

[Figure]

**Figure S4:** Monthly median and IQR (shading) diurnal patterns of storage heat flux ($\Delta Q_{S,ohm}$ and $\Delta Q_{S,ohm-rad}$) at Miyun (September 2012 to December 2013) (section 3.2). $\Delta Q_{S,ohm-rad}$: estimated $\Delta Q_s$ at MY by using land cover fraction of radiation source area.

Those large values of observed soil heat flux ($Q_G$) in Figure B2 that could not be reproduced by the OHM were all the maximum values of clear days, and they appeared at the same time as the maximum net radiation ($Q^*$). While in general, the maximum $Q_G$ of farmland appears later than that of the $Q^*$, for the fitted coefficient $a_2$ of farmland being negative in all seasons (Table B1). This phenomenon has been mentioned in previous studies, that is, OHM estimates perform satisfactorily in the mean but miss short-term variability (Grimmond and Oke, 1999; Roberts et al., 2006).

The above discussions have been added to Appendix B of the revised version (Line 708-712).

Text added (Line 708-712):

Some large observed $Q_G$ are not reproduced by the OHM model, as they are the maximum values and appear simultaneously with the peak of $Q^*$ on that day. Generally, the maximum $Q_G$ appears later than that of $Q^*$, for the fitted $a_2$ of farmland being negative in all seasons (Table B1). This phenomenon has been reported in previous studies, i.e., OHM estimates perform satisfactorily in the mean but miss short-term variability (Grimmond and Oke, 1999; Roberts et al., 2006).

Grimmond, C. S. B., and Oke, T. R.: Heat storage in urban areas: Local-scale observations and evaluation of a simple model, J. Appl. Meteorol., 38, 922–940, 1999.
Roberts, S. M., Oke, T. R., and Grimmond, C. S. B.: Comparison of four methods to estimate urban heat storage, J. Appl. Meteor. Climatol., 45, 1766–1781, doi:10.1175/JAM2432.1, 2006.

L279-83: I am not sure the comparison with the Swindon site is especially helpful, especially if no physical explanation is given for the difference. It would be much more interesting to use the Miyun dataset to investigate some of the variability with rainfall or radiation (e.g. as described in L290-2, L295).

The text of the comparison with Swindon site has been removed in the revised version (Line 346-351). The text (Line 380-387) and a plot (Fig. S3) of the comparison between daytime $\Delta Q_{S,ohm}$

and $\Delta Q_{s,res}$ within 24 h after rain in spring and summer has been added in the revised version to show that $\Delta Q_{s,ohm}$ is overestimated when sunny after rain.

Text added (Line 380-387):

At MY, it is common to see "sunny after rain" in spring and summer, which is similar to the pavement/road watering on sunny days. Pavement heat flux ($G$) is reduced after watering and the maximum value of $G$ is only half of that pavement without watering under the same sunny conditions (Hendel et al., 2015). However, the OHM coefficients of road/pavement/impervious used in this study were fitted from dry surfaces under clear sky conditions (Anandakumar, 1999; Doll et al., 1985; Wang et al., 2008), so it inevitably leads to an overestimation of $\Delta Q_{s,ohm}$ when sunny after rain. The comparison results of MY daytime $\Delta Q_{s,ohm}$ greater than $\Delta Q_{s,res}$ within 24 h after rain in spring and summer confirm this point (Fig. S3).

[Figure]

**Figure S3:** Daytime 30 min storage heat flux ($\Delta Q_{S,ohm}$ and $\Delta Q_{S,res}$) at Miyun by number of hours since rainfall (shape) and incoming radiation value (colour) in spring and summer. r2: coefficient of determination, N: evaluation data available.

L284-7: The second calculation of QS_RES_v2 as a lower limit is helpful and it would be nice to see this brought into the main text to make the OHM vs RES comparison more robust.

Done. Move Figure S5 into the main text, and now it is Figure 8 in the revised paper.

L287-290: It is not at all clear how this conclusion has been reached.

We have amended the statement that the uncounted biomass heat storage is the main explanation for the underestimated $\Delta Q_{s,ohm}$. The text has been modified as follows in the revised manuscript (L396-406).

Text added (L396-406):

This indicates the winter $\Delta Q_{s,ohm}$ underestimates are related to OHM coefficients for the prevailing wind directions. As cropland covers a greater proportion in the easterly wind direction, it suggests the wheat coefficients are poor in winter (Fig. B2d). This is because smaller fluxes are

obtained from using the soil heat flux ($Q_G$) data. These relations with $Q^*$ do not account for those components of heat storage fluxes above the soil heat flux plate, such as ground heat storage and biomass heat storage (Meyers and Hollinger, 2004; Oliphant et al., 2004). They constitute an important proportion of the total storage heat flux. Except for cropland and roof, OHM coefficients of other land cover types are fitted from the spring or summer observation dataset, while these coefficients may not be applicable to autumn and winter and cause the $\Delta Q_{s,ohm}$ to be underestimated in autumn and winter. Long-term datasets covering seasonal changes and surface conditions, especially winter, can provide more OHM coefficients for use (Anandakumar 1999, Ward et al. 2013).

L297-301: The sharp drop in QS_OHM that is seen in Fig 5b around sunset is seen all year round not just in summer (actually it seems to be even larger in winter than summer). This drop occurs for all the QS_OHM estimates but not for QS_RES. What could be the possible explanation for this and is it really likely to be a 'true' feature (as suggested by the text) or might it be an artefact of the estimation approach? I am not yet convinced. Was this drop seen in the observed soil heat flux data used in Appendix B? It could be worth checking the terms of the OHM equation individually to see what is producing this effect. If you used different timesteps for the OHM calculation does this feature still appear? The implied connection with the valley/mountain wind is not clear (L300). Similarly, what should the reader learn from the final sentence about clear/rainy conditions (L301)? Can you provide a physical explanation for this observation?

After reanalysis, it is found that a sharp drop in $\Delta Q_{s,ohm}$ is not a real feature, but a visual artifact given by the horizontal compression of the line plot.

Representative months (January, April, July, and October 2013) from spring, summer, autumn and winter are selected to show the monthly diurnal variation of net radiation $Q^*$ and $\Delta Q_{s,ohm}$. It can be seen from figure rev.3 that there is no sharp drop in $\Delta Q_{s,ohm}$ in any month, but the minimum value of $\Delta Q_{s,ohm}$ occurs in the evening (during 17:00-19:30). It is the inevitable result of the heat storage flux calculated by the OHM, which is mainly the joint action of the first and second terms of the OHM equation. So, the minimum value of $\Delta Q_{s,ohm}$ still occurs in the evening when use some different timesteps (such as 2 h, 3 h) for using OHM (Figure omitted).

Because the sharp drop is not true, the monthly diurnal variation of soil heat flux in farmland certainly does not present this phenomenon (Fig. rev 4).

The statement of sharp drop and relevant explanations for this phenomenon are deleted in the revised manuscript, including the analysis of the sharp drop in $\Delta Q_{s,ohm}$ caused by the decrease of net radiation due to the transformation of mountain-valley wind. Figure S4 of the monthly diurnal variation of soil and soil temperature difference in the submitted manuscript has also been deleted in the revised version.

[Figure]

Figure rev 3. Monthly mean diurnal patterns of net radiation ($Q^*$) and storage heat flux ($\Delta Q_{S,ohm}$) (a-d), and the three terms of the OHM equation (e-h) at Miyun in January (a, e), April (b, f), July (c, g) and October (d, h).

[Figure]

Figure rev 4. Monthly median and IQR (shading) diurnal patterns of net radiation ($Q^*$) (a) and soil heat flux ($G$) (b) at Weishan agricultural experiment station in 2009.

**Description of turbulent heat fluxes**

It is interesting to see the differences in seasonal variability compared to many of the previous sub/urban studies. However, more analysis is required to understand why this occurs. Are the seasonal patterns shown a result of the land cover (farmland versus urban) or the local climate (i.e. a rainy summer season)? More direct comparison with other studies in similar climates (e.g. Hong et al., 2020, the other Beijing studies mentioned in L53-55) would be very helpful in understanding the physical reasons for these observed differences and therefore transforming this study from only

locally relevant to being more widely useful for interpreting observations in other locations, improving process understanding, developing model parameterisations, etc.

Also, it is not clear whether these results essentially describe the farmland (i.e. would very similar results have been obtained if the flux station was installed in the fields?) or whether there are also urban effects (e.g. are the sensible heat fluxes enhanced by anthropogenic heat emissions?). It might be worthwhile reproducing Fig 6 using only data when the source area was over the farmland and (if there is enough data) also when the source area was over the more built-up area (similar to Fig 11, 12). In general the text could also give a clearer impression of the importance of the non-vegetated areas when the source area is to the east of the tower (perhaps adding site photos to an earlier section would help).

Compared with the previous reports of sub/urban sites and the observation results of two farmland (Fig. S5) with a similar climate to Miyun (Hong et al., 2021; Miao et al., 2012; Wang et al., 2015), it is confirmed that the seasonal patterns of the turbulent heat fluxes shown in Miyun are the result of the combined effects of local climate and land cover (farmland versus urban).

In previous sub/urban studies of the East Asian monsoon region, influenced by the climate characteristics of "hot-rainy summer, cold-dry winter", daytime $Q_E$ values are always greatest in summer and lowest in winter, respectively. Meanwhile, daytime $Q_H$ values are highest in spring or summer, which is closely related to the climate characteristics of spring. If spring is dry and little rainy, daytime $Q_H$ maxima occur in spring, such as at IAP, Beijing (Miao et al., 2012; Wang et al., 2015), Xianghe, Beijing (Wang et al., 2015), and Miyun, Beijing (this study), and Seoul Forest Park (Lee et al., 2021) and residential area (Hong et al., 2020) in Seoul; If spring is warm and humid, the daytime $Q_H$ peaks in summer, such as at Tokyo (Moriwaki and Kanda, 2004), Osaka (Ando and Ueyama, 2017) and Shanghai (Ao et al., 2016). However, daytime $Q_H$ values in all sites mentioned above are lowest in winter, except for Xianghe, Miyun, and Seoul Forest Park where daytime $Q_H$ values are lowest in summer. The common characteristic of these three sites is the presence of extensive irrigated cropland or irrigated forest within the flux source area. Notably, these characteristics are also different from irrigated farmland under the same climate background. The greatest daytime $Q_E$ values and lowest daytime $Q_H$ values of farmland appear simultaneously in spring when wheat grows (Fig. S5). The daytime $Q_H$ values of farmland peak in June, which is the wheat and maize rotation period.

To sum up, seasonal patterns of turbulent heat fluxes in Miyun are the result of the comprehensive effects of local climate and land cover (farmland vs urban).

Limit to insufficient data, it is unable to compare the seasonal patterns of $Q_H$ and $Q_E$ between farmland and built-up areas at Miyun directly. However, by comparing with the turbulent heat flux characteristics of farmland (Fig. S5), it could be concluded that the characteristics shown at Miyun are the joint action of farmland and urban effects.

Text added (Line 430-442):
In previous sub/urban studies of the East Asian monsoon region, daytime $Q_H$ values are highest in spring (IAP, Beijing, Miao et al., 2012; Wang et al., 2015; Xianghe, Beijing, Wang et al., 2015; residential area in Seoul, Hong et al., 2020; Seoul Forest Park, Seoul, Lee et al., 2021) or summer (Tokyo, Moriwaki and Kanda, 2004; Shanghai, Ao et al., 2016a; Osaka, Ando and Ueyama, 2017),

depending on whether spring is dry with little rain or warm humidity. Meanwhile, daytime $Q_H$ values are lowest in winter when the least solar radiation in the year occurs, except for Xianghe, Beijing and Seoul Forest Park, Seoul. Similar to MY, daytime $Q_H$ values of these two sites are lowest in summer, with irrigated cropland/forest within the flux source area. Hence at MY, the summer daytime $Q_H$ minima is attributed to the extensive irrigated cropland in the prevailing wind direction (Fig. 1c, d; 2d, h). This enhances available energy to support evaporation and transpiration and leads to smaller $Q_H$ values (Dou et al., 2019). Notably, these characteristics are different from wheat/maize rotation farmland under the same climate background. The daytime $Q_H$ values in farmland are lowest in spring, owing to sufficient irrigation and high-water consumption for wheat growth (Fig. S5).

[Figure]

**Figure S5:** Location of Weishan and Yucheng agricultural experiment station and Miyun (a) and monthly median and IQR (shading) diurnal patterns of net radiation ($Q^*$) (b, f), sensible heat flux ($Q_H$) (c, g), latent heat flux ($Q_E$) (d, h), soil heat flux ($G$) (e) at Weishan (b-e) and Yucheng (f-h) in 2009.

Ando, T. and Ueyama, M.: Surface energy exchange in a dense urban built-up area based on two-year eddy covariance measurements in Sakai, Japan, Urban Clim., 19, 155-169, doi: 10.1016/j.uclim.2017.01.005, 2017.

Hong, J. W., Lee, S. D., Lee, K., and Hong, J.: Seasonal variations in the surface energy and $CO_2$ flux over a high-rise, high-population, residential urban area in the East Asian monsoon region, Int. J. Climatol., 40, 4384–4407, doi:10.1002/joc.6463, 2020.

Lee, K. M., Hong, J. W., Kim, J. W., Jo, S. S., and Hong, J. Y.: Traces of urban forest in temperature and $CO_2$ signals in monsoon East Asia, Atmos. Chem. Phys., 21, 17833–17853, doi: 10.5194/acp-21-17833-2021, 2021.

Similar to the radiation section, there are a lot of statistics given here but not many new insights. Given the relevance of the farmland to the east, finer-scale temporal analysis linked to the state of the crops, irrigation schedule, soil moisture and meteorological conditions would be a useful addition. This seems to be the purpose of Section 3.4 but it is only done very broadly.

Miyun undergoes the transformation of mountain-valley wind every day, which means that the observation data representing the farmland characteristics are limited. In addition, we don't know the detailed irrigation schedule. Only by comparing the observed soil moisture of natural conditions and farmland three times a month, combined with rainfall data, we can infer whether irrigation was carried out in farmland during the two adjacent soil moisture observations. Thus, there is not enough data to develop a finer-scale temporal analysis linked to the state of the crops and irrigation schedule.
We try to provide a finer-scale temporal analysis in Section 3.5, to present the effects of plant growth, rainfall, and irrigation on energy partitioning through Figures 14 and 15.

L358-64: This simple comparison with two very different sites is not particularly useful.

Given the few reports on the monthly mean ratios of $Q_H$ and $Q_E$ to $Q^*$ for the whole year, we retain the sentences about comparing the ratios between Miyun, Tokyo, and Oberhausen (Line 493-498).

L381-2: This comparison is much more interesting and perhaps could be further developed by concisely incorporating the relevant information from the references so that the comparison is clearly extended beyond Tokyo and Oberhausen. Similarly in L412-6 – it is difficult for the reader to get the importance of the message if they have to go and look in references and basically make the comparison themselves.

Figure 12 is added to show the summer and winter Bowen ratios of Miyun and other suburban sites in the world in the revised manuscript. Similarly, figure 13 has also been added to show winter Bowen ratios at Miyun and urban sites with vegetation coverage of 1-31% in the revised version.

[Figure]

**Figure 12:** Daytime (the definition of daytime depending on which is given in the corresponding publication) Bowen ratio during summer (a) and winter (b) versus vegetation fraction (λ𝑣) for MY and for various suburban sites in the literature (see the legend for references).

[Figure]

**Figure 13:** As Fig. 12 but winter daytime Bowen ratio for MY and for various urban sites. At Bi09, Ba02u1, Ba02u2, Sh12, Bowen ratio is the mean value of Dec-Feb; At My13, Me93 and Mo07u, is the mean value of Dec; At Ob10u, is Jan; At Ou03 and TK01, is Feb. All Bowen ratios are accurately provided in the paper, except for Sh12 extracted from the plot and Mo07u extracted from ratios of $Q_H/(Q^* + Q_F)$ and $Q_E/(Q^* + Q_F)$.

L408-428: These paragraphs should really form the main contribution of the manuscript, where the reader learns how these factors relate to the observations set out in previous sections. However, there is hardly any analysis here.

Fig 9 and 10 are briefly mentioned without much interpretation. The meaning of Fig 9 is unclear to me (not helped by the confusing symbol and colour choices). What is the reader supposed to compare with what? Adding two or three sentences summary in the text could be helpful, although these data don't appear to show a consistent pattern. Since these data are at a different timescale to the monthly fluxes, it is very difficult for the reader to see the connections briefly mentioned in the text.

Figure 9 (ordered as Figure 14 in the revised manuscript) has been redone to show the difference in soil water content between irrigated cropland and natural conditions more clearly. Text has been added to explain that irrigation will affect the flux values and energy partitioning (Line 549-553).

Text added (Line 549-553):

As shown in Figure 14, the soil moisture of natural conditions and cropland were almost the same at the end of April (28th April). With only 3 days (26th, 27th and 28th May of 0.1, 0.4 and 0.2 mm, respectively) having rain in May 2013 (Fig. 2b), the soil moisture of natural conditions decreases dramatically in May, whereas of cropland increases in contrast. This indicates that cropland has irrigation to replenish water.

[Figure]

**Figure 14:** Gravimetric soil moisture (%) at a depth of 0.1 m measured on the 8th, 18th and 28th of each month at the MY site from 18 March to 18 November, 2013 under natural condition and irrigated cropland. The shading is May and July 2013, respectively.

Despite crop growth being relevant (L423) I don't see this information presented anywhere.

Description of crop growth cycle has been added in Line 545-549 of the revised version.

Text added (Line 545-549):

Winter wheat and summer maize are the two predominant crops and cultivated in rotation. The growing season of winter wheat is from October to mid-June, while maize is planted in late June and harvested in the end of September every year. June and October are the intermittent months for the two crops. Irrigation is frequent during wheat growth in spring, because of the drought and little rain climatic conditions.

Similarly Fig 10 is referred to in single sentence (L421) and the reader has to do all the interpretation. Consider combining some of Figs 7-11 and adding additional information (such as crop growth and harvesting dates, irrigation times) or averaging the observations over relevant time periods so that the reader can see that connections suggested in the text are supported by the data – i.e. can see how the surface fluxes respond to the supposed controls.

The interpretation for Figure 10 (Figure 15 in the revised manuscript) has been added in the revised version (Line 553-559), to provide a clearer explanation of the impact of irrigation on energy partitioning.

Text added (Line 553-559):

The May $\beta$ clearly presents a decreasing tendency even if there is more than 20 days of no rain (Fig. 15), as external water provided by cropland irrigation makes the available energy favor the latent heat flux. The monthly mean $\beta$ of 0.50 in May (Fig. 11), clearly indicates the importance of this additional source of water.

L429-35: This short paragraph mentions two figures but offers no analysis. Fig 11 is useful and relevant and should be analysed in more detail in the text. Why are the fluxes normalised by Q_down here and not Q* as above? Are similar or different patterns seen if Q_down is used for normalisation in Fig 7 or Q* is used for normalisation in Fig 11. Might it be worthwhile to add another panel for QS to Fig 11 (e.g. to help provide evidence for or against the hypothesis in L378-9)? The division into wind sectors needs to be properly introduced (not just mentioned in the caption of Fig 12).

One of this paper's objectives is to analyze the energy partitioning characteristics of $Q^*$ between $Q_H$ and $Q_E$ and confirm the factors affecting the energy budget. That incoming radiation $Q_\downarrow$ is selected as normalization in Figure 11 (Figure 16 in the revised version) because we want to discuss the impact of land cover/use types on energy partitioning. The $Q^*$, as the difference between the incoming and the outgoing radiation, is affected by the land cover/use types while using $Q_\downarrow$ can avoid this problem.

As Figure rev5 shown, it is similar to the energy partitioning pattern of $Q^*$ if $Q_\downarrow$ is used for normalization in Figure 7 (Figure 10 in the revised version). From November to April $Q_H$ dominates the $Q_\downarrow$ ratio, whereas from June to September $Q_E$ dominates the $Q_\downarrow$ ratio. For May and October, they have a similar ratio with $Q_\downarrow$ (Fig. rev 5). However, the largest $Q_H/Q_\downarrow$ does not appear in winter like $Q_H/Q^*$, but in spring. This is because the snow in winter leading to a smaller $Q^*$, which result in larger $Q_H/Q^*$.

[Figure]

[Figure]

Figure rev 5. Monthly median, IQR and mean (30 min) of daytime ($K_↓$> 5 W m$^{-2}$) sensible heat flux $Q_H$ and latent heat fluxes $Q_E$ normalized by net all-wave radiation $Q^*$ (a) and incoming radiation $Q_↓$ (b) at MY (September 2012 to December 2013).

Similarly, if the $Q^*$ is used as normalization in Figure 11 (Figure 16 in the revised manuscript), the difference in energy partitioning between building- and farmland-dominant sectors does not change, which is the same as the result of using $Q_↓$ as normalization (Fig. rev 6).

[Figure]

Figure rev 6. Median, IQR and mean flux ratio (30 min, number of periods indicated) by season when wind is from the building- (red) and farmland-dominant directions (dark green) (a) $Q_H/Q^*$ (b) $Q_E/Q^*$.

The heat storage flux ($\Delta Q_S$) of Tokyo (Moriwaki and Kanda, 2004), IAP Beijing (Miao et al., 2012) and Mexico City (Oke et al., 1999) cited in lines 373-378 of the submitted manuscript is determined as the energy balance residual from direct observation of net all-wave radiation ($Q^*$), sensible heat ($Q_H$), and latent heat ($Q_E$) fluxes, i.e., $\Delta Q_S = Q^* - Q_H - Q_E$. If the $\Delta Q_S$ of Miyun is calculated by the same method, Miyun daytime $\Delta Q_S$ was 25-53% of $Q^*$ in winter, which was lower than that of Tokyo (0.60-0.62; Moriwaki and Kanda, 2004), Mexico City (0.58; Oke et al., 1999), and also lower than Miyun $Q_H/Q^*$ of 0.41-0.57. So, it is expressed in the submitted manuscript (Line 378-380) that the $\Delta Q_S$ is a small proportion of in the $Q^*$ energy partitioning.

To avoid confusion, the calculation formula for the heat storage flux (here $\Delta Q_S = Q^* - Q_H - Q_E$) and $\Delta Q_S/Q^*$ ratios of 0.25-0.53 are added in the revised version (Line 499-501).

The data ($Q_↓, \beta$) used to show the energy partitioning characteristics in Figure 11 (Figure 16 in the revised version) are all from direct observation ($Q_H, Q_E$), or the sum ($Q_↓$) or ratio ($\beta$) of these direct observation values. While $\Delta Q_S$ is calculated as residual term of energy balance equation and its uncertainty is relatively large. So, the panel for $\Delta Q_S$ does not been added in Figure 11 (Figure 16 in the revised version).

In addition, the data from December 2012 was accidentally omitted when plotting Figure 11 in the submitted manuscript, and the amount of data involved in winter statistics was corrected and Figure 11 (Figure 16 in the revised version) was redrawn.

The division into the building- and farmland-dominant directions has been added to Lines 574-579 in the revised manuscript.

Text added (Lines 574-579):

As the surface of the daily averaged fluxes source area is mainly covered by vegetation (60%) and impervious (87%) at 30-150° and 210-360° sectors of the tower respectively, and farmland and

buildings are separately the most representative land use/cover types in the two sectors, these two sectors are selected to compare energy fluxes ratios and referred to as farmland- and building-dominant directions accordingly.

A more detailed analysis of Figure 11 (Figure 16 in the revised version) has been added to Lines 581-585 in the revised manuscript.

Text added (Line 581-585):
Compared to impervious surfaces, the cropland surface obviously transforms between "dry" and "wet" more frequently and with longer transition time, due to cropland irrigation and water storage by soil. This results in a greater amplitude of $Q_H/Q_\downarrow$ and $Q_E/Q_\downarrow$ from the cropland-dominated direction. The inter-quartile range and the differences between the mean and median of ratios are clearly greater from the cropland direction (Fig. 16).

It's not clear what the reader is supposed to learn from Fig 12. Don't these two plots basically suggest very little dependence of Bowen ratio on time since last rainfall and little difference between the two sectors? The text L433-5 doesn't really make sense and is more of a hypothesis which does not seem to be convincingly supported by the data.

For the building-dominant direction, the frequency of $\beta$ between 0-1 did not exceed 30% when more than 24 hours since the last rainfall, while for the farmland-dominant sector, there is still about 35% $\beta$ between 0-0.5 on the fourth day (196 hours) since the last rainfall. This indicates that land use/cover have an impact on energy budget.

In addition, the frequency of $\beta$ between 0.5-1 increases from 35% to about 60% at the 14th-16th hours since the last rainfall for the farmland-dominant direction, which suggests that cropland irrigation supplementing water is also one of the factors affecting energy partitioning.

Figure 12 (Figure 17 in the revised version) has been adjusted in the revised version and the labels in days on the x-axis that are more than 24 hours since the last rainfall are marked in red to show the difference.

[Figure]

**Figure 17:** Frequency (F, %) of Bowen ratio $\beta$ values by time since last rainfall for (a) building-dominant (210-360°) and (b) farmland-dominant (30-150°) directions.

**Conclusions need to offer something more than a repetition of results**

The conclusion section is simply a repetition of the results. Too many statistics are given here and there is no summary, synthesis or bigger picture. The final paragraph refers to the importance of these kind of observational studies, but unless the previous sections can be further developed to provide insight into these key factors (i.e. to improve understanding of processes) this study cannot claim to be useful for any of the purposes given in L478-81.

The previous sections have been made significant modified. The results could confirm the combined importance of precipitation, irrigation, crop/vegetation growth and land cover as being key factors affecting energy partitioning in Miyun. We hope that study results enhance the understanding of the surface-atmosphere energy exchange over sub/urban sites.

**Specific comments – minor**

L46: Would be good to point out that this extreme example from Basel is a car park site, and not a typical urban neighbourhood site.

It has been noted in the revised manuscript that this observation site from Basel is a car park site (Line 47).

L53-60: It's not very clear to the reader which of the sites are being referred to as 'urban' and 'suburban' here. Then in L60, it's not clear if the suburban site (Miyun) is the same as introduced above or a new site that has not previously been mentioned in the text. Please rephrase this paragraph to avoid confusion.

Using the Local Climate Zones (LCZ) classification (Stewart and Oke, 2012), LCZs 1-4 sites are referred to as 'urban' and LCZs 5-9 as 'suburban' in this study.

The definition of urban and suburban sites is added in the revised manuscript to avoid confusion (Line 59-61).
Text added (Line 59-61):
According to the Local Climate Zones (LCZs) classification (Stewart and Oke, 2012), LCZs 1-4 sites are defined as 'urban' and LCZs 5-9 as 'suburban' sites in this study. If surface is covered by more than 80% vegetation (including trees, grass, crops etc.), this observation station is referred to as rural site.

Fig 1: The aerial image in (b) and land cover in (c) don't seem to match up very well close to the tower – it looks as though the tower position is 100-150 m further north in (b) compared to (c) and possibly on a building in (b) but on grassland in (c). It would be good to state in the text the land cover directly at the flux station (was the tower mounted at ground level?) and for the nearby weather station. It would also be helpful to add the weather station to the maps in Fig 1. The colour bar in (a) needs an axis label and units (or a more complete description in the figure caption). Using consistent colours to represent the land cover in panels (c) and (d) would help the reader.

The position of the tower in Fig 1c has been adjusted to match that in Fig. 1b.

The tower was mounted at the ground level directly. The description of the land cover at the flux station has been added to the revised manuscript (Line 80-82).

The position of the weather automatic station is the position of the tower marked in Fig. 1c, and the legend has been modified.

The color bar in Fig. 1a shows the elevation and unit is meter. The text "Elevation (m)" has been added in Fig. 1a.

Fig. 1d was redrawn so that its color was consistent with Fig. 1c.

[Figure]

**Figure 1:** Study area (a) Beijing topography with Miyun (MY) (inset; location in China, red dot); (b) aerial view around MY flux tower (Google Earth 2017); (c) land cover within 1 km radius around MY flux tower (black dot) and 90% eddy covariance source area for daytime ($K_\downarrow$> 5 W m$^{-2}$, black line), night (blue) and daily average (pink), and (d) daily mean land cover derived from GF-2 High-resolution image (CCRSDA, 2016) for 30° wind sectors of the 90% source area (shown in c).

L72-77: A lot of heights, distances and directions are given here and it is not easy for the reader to get a clear impression of what is important for this site or match the descriptions to the aerial imagery or land cover map in Fig 1. If the building heights and distances in each direction are very relevant, consider adding a building height map as an extra panel in Fig 1; if this information is not especially relevant (which seems to be true) then less detail can be given here. In any case, the relevant building height information should be consolidated with similar information given in Section 2.4. The distance to the tallest building should be mentioned. It might also be helpful to show where each of the crops are (perhaps in the extra panel in Fig 1) as the reader has to guess where the wheat and maize mentioned in L166 are growing.

This paragraph has been rewritten, and the simplified text is on lines 176-179 of the revised version. The distance (380 m) to the tallest building is added to line 179.

The light-yellow area in Fig 1c represents farmland, where crops grow. The farmland at the observation site is under a wheat/maize rotation system. The growing season of maize is from the end of June to the end of September every year, and the rest is wheat. The detailed growth time of wheat and maize is supplemented in Section 3.5 (line 545-549) of the revised version.

L170: Are these respective anemometric values for z0 and zd? It is not completely clear from the text.

Yes, $z_0$ and $z_d$ values are 2.9 m and 4.1m, respectively.

The words "$(z_0)$" and "$(z_d)$" have been added after the "2.9 m" and "4.1 m" in Line 187 to explain the values of $z_0$ and $z_d$ more clearly.

L181-4: These seem unrealistically precise distances for a footprint model which can only provide an indication of the likely source area. Probably these should only be given to two significant figures at most.

Given the clear differences in footprint length and corresponding land cover fraction between the daytime and night, the original text is not modified.

L201-2: It's not clear which definition has been used for daytime/night-time – the incoming shortwave radiation or the times given.

The daytime is defined as $K_{\downarrow} > 5$ W m$^{-2}$. That different daytime period of $K_{\downarrow} > 5$ W m$^{-2}$ in each season is due to the seasonal mean diurnal variation of $K_{\downarrow}$ patterns being different from each other.

To reduce ambiguity, Section 2.7 is revised as follows in the revised manuscript (Line 219-229):
Analysis is done by season (spring: March–May (MAM), summer: JJA, autumn: SON, and winter: DJF) and by time of day (daytime: $K_{\downarrow} > 5$ W m$^{-2}$, night: other times).
Missing data are not gap-filled. The mean values of radiation and turbulent fluxes for each half-hour during a day within the month (season) are first calculated to get monthly (seasonal) mean diurnal patterns. Then the monthly (seasonal) daytime ($K_{\downarrow} > 5$ W m$^{-2}$) or daily (24 h) mean values are averaged from corresponding periods within the mean diurnal patterns. The monthly (seasonal) daytime (daily) mean ratios are the ratios of daytime (daily) mean values of corresponding radiation and energy fluxes.

Table 1 is very difficult to interpret, the caption is difficult to follow and I'm not sure the information is really necessary in this much detail. Also the table does not include the information suggested by the text concerning quality control procedures (L102-3) or definitions (L445-6).

Table 1 has been deleted from the revised manuscript. The definitions in the header of Table 1 are moved into in the header of Table 2 (Table 2 in the submission has been reordered as Table 1 in the revised version).
The amount of the deleted $Q_H$ and $Q_E$ data due to poor quality, during rain and two hours after rain has been shown on line 109-110 of revised version.

L267-70: It's not clear to me how this conclusion was reached from looking at Fig 5a. What is the calculated annual gain/loss?

The annual total of heat storage flux is the sum of the monthly total of 12 months in the year. The monthly total value is the product of the monthly daily total value and the number of days in the month.

The yearly total value of $\Delta Q_{s,ohm}$ is 352.8 MJ m$^{-2}$ in 2013. By calculating the yearly total value, we found that $\Delta Q_{s,ohm}$ in Miyun was overestimated in spring and summer. The reason is that the OHM parameters of road/pavement/impervious were fitted at dry surfaces, under clear sky conditions (Anandakumar, 1999; Doll et al., 1985; Wang et al., 2008). While pavement heat flux ($G$) is reduced after watering and the maximum value of $G$ is only half of that pavement without watering in sunny days (Hendel et al., 2015). At Miyun, it is common to see "sunny after rain" in spring and summer, which is similar to the pavement/road watering on sunny days. So, the $\Delta Q_{s,ohm}$ will be overestimated under these conditions because it only account for the net radiation value according to the OHM equation. The comparison results of MY daytime $\Delta Q_{s,ohm}$ greater than $\Delta Q_{s,res}$ within 24 h after rain in spring and summer confirm this point (Fig. S3).

As for the $\Delta Q_{s,ohm}$ is underestimated in winter, it is because monthly daytime and daily total values of $\Delta Q_{s,ohm}$ are lower than those of $\Delta Q_{s,res-v2}$.

In the revised manuscript, the section 3.2 about heat storage flux has been rewritten, the analysis of $\Delta Q_{s,ohm}$ overestimation in spring and summer in Miyun (Line 375-387) are added, and the text (Line 391-406) are also adjusted to explain that $\Delta Q_{s,ohm}$ is underestimated in winter.

Anandakumar, K.: A study on the partition of net radiation into heat fluxes on a dry asphalt surface, Atmos. Environ., 33, 3911–3918, doi:10.1016/S1352-2310(99)00133-8, 1999.

Doll, D., Ching, J. K. S., and Kaneshiro, J.: Parameterization of subsurface heating for soil and concrete using net radiation data., Bound.-Layer Meteorol., 32, 351–372. doi:10.1007/BF00122000, 1985.

Hendel, Martin., Colombert, Morgane., Diab, Youssef., Royon, Laurent.: An analysis of pavement heat flux to optimize the water efficiency of a pavement-watering method. Applied Thermal Engineering, 78, 658-669, doi: 10.1016/j.applthermaleng.2014.11.060, 2015.

Wang, C. G., Sun, J. N., and Jiang, W. M.: Observation and analysis on thermodynamic characteristics of heat storage of urban roof, Acta Energiae Solaris Sinica, 6, 694–699, doi:10.3321/j.issn:0254-0096.2008.06.011, 2008.

L270-72: The meaning here is unclear - the text needs to be rephrased and possibly further explanation added. It' not clear what this conclusion about wind direction dependence is based on.

The sentences of line 270-272 in the submitted manuscript have been deleted. The reason for the $\Delta Q_{s,ohm}$ underestimated in winter has been explained in the revised manuscript (Line 391-406).

L273-5: Where are the mean and median values shown?

We want to express that the impacts of the differences in data availability on comparison between $\Delta Q_{s,ohm}$ and $\Delta Q_{s,res}$ during winter are negligible.

When data availability of $\Delta Q_{s,ohm}$ and $\Delta Q_{s,res}$ is different (N= 3901 for $\Delta Q_{s,ohm}$ and 3419 for $\Delta Q_{s,res}$, respectively), the median and mean for $\Delta Q_{s,ohm}$ in winter (Dec-Feb) are -41.3 W/m$^2$ and -22.2 W/m$^2$, for $\Delta Q_{s,res}$ are -16.7 W/m$^2$ and 12.5 W/m$^2$.

When data availability is the same (N=3417), the median and mean for $\Delta Q_{s,ohm}$ are -41.7 W/m$^2$ and -22.0 W/m$^2$, for $\Delta Q_{s,res}$ are -16.8 W/m$^2$ and 12.4 W/m$^2$.

To avoid ambiguity, this sentence has been deleted from the revised manuscript.

L363: How was this vegetation fraction of 34% calculated? Is it a source-area averaged value?

It is the land cover fraction of vegetation within the 90% eddy covariance source area for the daily average during the observation period. That is the proportion of vegetation within the pink outline in Figure 1 (c). The vegetation includes trees, grassland, and farmland.

L533-8: It's not clear in this paragraph which population data has been used for the QF estimation and what the implications for the analysis are.

The total population of Miyun District in 2012 and 2013 was 545000 and 548000 respectively, and the total area of the district was 2229.45 km$^2$, so the population density used in Line 533-538 were 245 people km$^{-2}$ in 2012 and 246 people km$^{-2}$ in 2013, respectively.

These population density data have been added to the Figure A1 caption (Line 665-666) and Line 673-374 in the revised manuscript.

**Technical corrections/language edits (not exhaustive!)**

L22: suggest changing to 'effects of climate change' since 'urban' is already mentioned several times in this sentence

Done. Changed to "effects of climate change".

L31: It's not clear what 'These' refers to – please rephrase explicitly.

"These" has been specified as "These measurements".

L35-6: Again it's not clear what 'these' refers to – consider rephrasing so that the start of the next sentence ('The EC method sites…') is also better expressed

The sentence "In recent years, these have been..." is modified to "Since the 1990s, EC methods are gradually undertaken in cities around the world" and placed before the sentence "These measurements have provided information about surface energy exchanges". (Line 31)

L41: an 'and' appears to be missing here

Done. The word 'and' has been added (Line 42).

L42-44: This sentence does not make sense. Please rephrase

This sentence has been deleted in the revised paper.

L44: 'The ratio of impervious…'

Done. 'The impervious…' changed to 'The ratio of impervious…'. (Line 45)

L56: I don't understand what is meant by 'both individually and concurrently'

The original sentence is modified to "These include observations for 1-year at the urban site individually, and at the urban and rural sites concurrently." in the revised manuscript (Line 64-65).

L60-1: This sentence does not quite make sense. Suggest rephrasing as 'We focus on the impact of site characteristics and environmental factors on energy partitioning.' However, 'environmental factors' is quite vague so it would be good to be more explicit here.

The sentence was modified to "We focus on the impact of site characteristics and precipitation on energy partitioning." (Line 70-71)
'Environmental factors' are replaced by 'precipitation' in the revised version.

L87: Suggest rephrasing as 'and 90% eddy covariance source area'

Done. The sentence has been rephrased as 'and 90% eddy covariance source area'. (Line 94)

L103-4: 'the anemometer was installed with respect to magnetic north'

Done. (Line 111-112)

L111: I think you only need to mention the month-long gap here (the other periods of 1-2 days are probably not important).

The description of data missing of 1-2 days has been deleted in revised manuscript.

L129: There is no bare soil fraction mentioned in the land cover description or shown in Fig 1(d).

Figure 1d has been modified and legend of bare soil has been deleted.

L138: Delete 'The'

Done (Line 150).

L144-8: This doesn't make sense grammatically – please rephrase.

The text has been revised as follows in the revised version (Line 156-159):

Easterly winds (30-120º) prevailed at night and the whole day for every season in MY, with a frequency of 50% and 38% in spring, 54% and 43% in summer, 69% and 54% in autumn, and 72% and 62% in winter, respectively (Fig. 2c-j). During the daytime, wind also came from the southwest direction (180-270º) because of the mountain-valley breeze, despite the southwest wind differed in

the start and end times among seasons (Fig. 4g-j).

L159-62: This is difficult to read and doesn't quite make sense – please rephrase

Text has been modified as follows (Line 176-179 in revised version):
The buildings vary from 3.0 to 50.4 m. The farmers' houses range from 3.5-7.3 m to the northeast of the tower. The residential buildings (6 floors) are to the west, northwest, and north of the tower, with consistent heights of 16.0-18.5 m. The height of buildings to the south, southwest, and north of the tower vary greatly, but most of the buildings exceed 20 m.

L191: 'difficulty'

Done. Wrong spelling has been corrected.

L191: 'use' or 'used'

Changed to 'use'.

L194-6: Please rephrase

This sentence has been rewritten as follows (Line 213-217 in revised version):
(2) Energy balance residual $\Delta Q_{s,res} = (Q^* + Q_F) - (Q_H + Q_E)$ : includes the uncertainty of other terms in the equation. On the one hand, the EC fluxes ($Q_H$ and $Q_E$) underestimate by 10-20% (Wilson et al., 2002; Foken et al., 2008), and on the other hand, given the prevailing easterly winds (30-150º) (Fig. 2c-j), vegetation accounts for 60% of the area of which 44% is cropland (Fig. 1c, d) if $Q_F$ is included, the $\Delta Q_{s,res}$ values should be regarded as the upper limit of heat storage flux (Ward et al., 2013).

L204-5: I don't understand the meaning here. Please rephrase.

This sentence has been rewritten as follows (Line 223-229 in revised version):
Missing data are not gap-filled. The mean values of radiation and turbulent fluxes for each half-hour during a day in the month (season) are first calculated to get their mean diurnal patterns. Then the daytime ($K_\downarrow > 5$ W m$^{-2}$) or daily (24 h) mean values are averaged from corresponding periods within the mean diurnal patterns. The daytime (daily) mean ratios are the ratios of daytime (daily) mean values of corresponding radiation and energy fluxes.

L220: 'becomes increases' à 'to increase', 'decreases' à 'decrease'

Done. Change to "becomes an increase to 0.6 and then a decrease with days since snowfall".

L223: Is 'variability' really meant here, i.e. are the statistics given a measure of variability?

This 'variability' refers to the range of midday albedo.

To avoid ambiguity, the original sentence is modified to "During the spring, summer, and autumn daily midday (10:00-14:00) albedo is between 0.128 and 0.209 in 2012 (0.113-0.209, 2013),

while in winter in 2012 (2013) it reaches 0.155-0.478 (0.134-0.293) with the variations in snow cover." in the revised version (Line 256-258).

L386-7: This sentence doesn't belong here without a '(see Section 3.5)' or similar.

A '(see Section 3.5)' has been added to the Line 508.

L453: I'm not sure it makes sense to call Q* a component of the radiation balance.

The sentence has been changed to "All components of the radiation balance and net radiation have unimodal annual patterns" (Line 600).

**References**

Hong J-W, Lee S-D, Lee K, Hong J (2020) Seasonal variations in the surface energy and CO2 flux over a high-rise, high-population, residential urban area in the East Asian monsoon region. International Journal of Climatology 40: 4384-4407 https://doi.org/https://doi.org/10.1002/joc.6463

A citation to the SEB study by Hong et al (2020) has been added in the revised manuscript (Line 42 and 432).

---

## Author Comment (AC3)

We would like to thank the reviewers for their comments.

- Our point-by-point responses are in *blue*
- Our proposed changes to the manuscript in *red*.   Line numbers (L) are the new ones unless otherwise indicated.

**Reviewer 2**

Review of: Title: Surface energy balance fluxes in a suburban of Beijing: energy partitioning variability
Authors: Dou, J., S. Grimmond, S. Miao, et al

Summary:
========
This manuscript describes observations of radiative and turbulent fluxes over 16 months in a suburb of Beijing. The paper describes the results and fills in any missing information with models and/or ancillary information.  As one would expect, as plants/vegetation starts transpiring and there is liquid water for evaporation in the summer, the latent heat flux increases at the expense of the sensible heat flux.  There is some nice additional information related to how irrigation might be affecting these results...however, this type of information/result is not anything new or unexpected.  I am not an urban specialist (and some of the comparisons with results from other urban sites seems a bit difficult to appreciate).

In general, the manuscript is well-written, though there are a few grammatical errors and confusing sentences which I highlight below.

There is a lot of information in the manuscript and the presentation is done well, but I don't think the overall results about how Qe increases at the expense of Qh are highly novel.  I also think the manuscript would be much more impactful if the storage terms are actually measured rather than roughly estimated with a model and then hand-waving about the results look "reasonable".  Finally, some of the references related to the SEB closure are a bit old/outdated and a more recent summary of SEB work should be provided.

   We would like to thank the referee for the helpful comments and suggestions, as well as some interesting new references.

   We have compared and summarized the previous reports of sub/urban sites and the observation results of two farmland with a similar climate to Miyun and believe that the seasonal variation characteristics of turbulent flux in Miyun are the result of the combined effect of local climate background and large-scale farmland irrigation. These new analyses have been added to Lines 430-442 in Section 3.3 of the revised manuscript.

   Regarding the issue of heat storage terms and SEB work, please refer to our specific response below.

General Comments:
==================

1. There is something I don't understand about the footprint analysis...the primary wind direction is from NE/E (ie, Fig 2c-2f). however, the flux footprint (shown in Fig.1c) is from the E/SE....shouldn't the footprint follow the wind direction?  It also does not make sense to me that the shape of the

footprint in Fig.1c is as "round" as it is...I would expect a very small contribution from the NW direction (since the wind rarely comes from that direction).  What am I not understanding about this?

The eddy covariance flux footprint provides a basis to identify which area (and weighting) should be used to estimate the land surface fractions impacting the measurements. Generally, there are two methods for determining the flux footprint, the one way determine the footprint and resulting land cover fractions dynamically (e.g. for each 30-min period), whereas the other assumed a constant radius/contour (e.g. based on a climatological analysis or rule of thumb).

This study used the latter method. At Miyun, footprint distance (Fig. rev 1) is calculated by using the method of Kljun et al. (2004) in Eddypro software (v6.1.0 beta, LI-COR). After obtaining the output results of footprint distance (m) each 30-min during the observation period, the mean values of footprint distance every 5º are calculated according to wind direction. This calculated footprint distance of 360º around flux tower is recorded as a diurnal average. The daytime is defined as incoming short-wave radiation $K_\downarrow > 5$ W m$^{-2}$. Taking this as the standard, the footprint distance data involved in the calculation are divided into two data sets: daytime and nighttime. Then, according to the calculation process mentioned above, daytime and night averaged footprint distances are obtained (as Fig. 1c shown), respectively.

[Figure]

Figure rev 1. Diagram of flux source areas (90%, shaded ovals) and footprint distances (black lines) for 30-min period in various wind directions by using the flux source area model of Kljun et al. (2004) for study site.

2. In order to improve/push the science, there should be some measurements of the storage terms (rather than just state that these are "too difficult", so we are going to use a model).  In my experience, the storage terms are critical for getting a successful surface energy budget closure (e.g, Leuning et al 2012; Swenson et al 2018).  I realize that it's too late to change this, but I feel the paper would be a more significant contribution if those terms were measured.

We agree with your suggestion, however, we did not have the ability to conduct this measurement at that time. To carry out heat storage terms observation in urban, it is the basic experimental requirement to install a lot of thermometers in different land cover types (building roofs and walls, roads, grassland, cropland, wood, etc.) to measure surface temperature. Our research funds and the number of instruments cannot meet the experimental requirement. In addition, it is difficult to find suitable buildings and cropland for instrument installation. We regret not doing this.

3. Related to the previous comment: how is the thermal heat storage by buildings taken into account? It looks like values from Nanjing are used (Table B1). Were any measurements of building temperatures taken to confirm this? An IR temperature sensor? Also, though the soil heat flux is considered, the heat stored in the soil layer above the soil flux sensor does not seem to be considered. This can be a significant contribution to the surface energy budget.

One of the methods to estimate the heat storage flux is using the OHM model in this study. The urban canopy layer is viewed as a "box" by enclosing all surface elements (ground, buildings, and vegetation), and the heat storage is estimated by the accumulation of the heat storage of all surface elements in the box according to the proportion of their area as a weight.

The parameters in the OHM model used for the calculation of the heat storage by buildings at Miyun are the values from Nanjing (Wang et al., 2008). Measurement of building temperature is not carried out in this study.

To calculate the heat storage above the soil heat flux plate, it needs to use the observation data of soil temperature and volumetric water content and need to know the soil bulk density. Unfortunately, we don't have these data, so cannot calculate it.

Specific Comments
===================

* The authors suggest that the title should be: "Surface energy balance fluxes in a suburban area of Beijing: energy partitioning variability. I would suggest something shorter/simpler: "Surface energy fluxes in a Beijing suburb: energy partitioning variability.

Thank you for your suggestion. Since the energy flux terms analyzed here are all components of the energy balance equation, we feel that the current title is more appropriate and did not modify it.

* l.31, "information about surface energy balance exchanges..." I would re-word as, "information about surface energy exchanges..."

Done. The word "balance" in the sentence (Line 32) has been deleted.

* l.35, "In recent years, these have been...". Specify what you mean by "these"..."these measurements"? If so, cite a few cities and references as examples in recent years? The examples you list on l.28-29 are from 2005 which I don't consider "recent"...

The sentences "In recent years, these have been..." is modified to "Since the 1990s, EC methods have gradually been undertaken in cities around the world. These measurements provide information about surface energy exchanges, helped development of parameterizations (Grimmond and Oke, 1999b, 2002; Järvi et al., 2019), and evaluation and application of land surface models (Grimmond et al., 2010; Järvi et al., 2011; Järvi et al., 2014; Karsisto et al., 2015; Ward et al., 2016; Liu et al., 2017; Kim et al., 2019) and remote sensing products (Kim and Kwon, 2019)." in the revised version (Line 31-36).

"These" has been specified as "These measurements" (Line 31-32).

* l.43, I don't understand the sentence that starts with "These have located..."?

Sentence deleted (Line 43-45).

* l.52, "..provide many new insights." is vague. can you mention/highlight a few of the most important new insights from all of these papers/studies?

A summary of the new findings and new insights of two EC long-term observation studies has been added to line 53-58.

Text added (Line 53-58):
Such as at a suburban site in the UK, energy partitioning favors turbulent sensible heat during summer but latent heat in winter and is strongly dependent on land cover fractions (Ward et al., 2013). However, the seasonal variability of energy fluxes normalized by net radiation is relatively small in a residential neighborhood of Singapore, as the measurement site in the equatorial with a very small variability in the background climate (Roth et al., 2017).

* l.94, is "lateral" wind the same as cross-wind? If so, isn't lateral wind a component of the horizontal wind?

Yes, lateral wind is the same as crosswind. The words "horizontal, and lateral" has been changed to "along-wind, and crosswind" in revised manuscript (Line 101).

* l.100, why do you use the double-coordinate rotation? Why not the planar fit? Does this choice affect/change the results?

We chose double coordinate rotation instead of planar fit rotation for two reasons.
Firstly, planar fit correction is not suitable for Miyun. The effect of planar fitting on fluxes estimate was ever considered over an urban landscape in Helsinki by Vesala et al (2008). They found that the vertical rotation angle (the angle that indicates the slope of the surrounding surface) is a function of the wind direction, and the angle values were not in agreement with the local topography in the sector where massive buildings are located (the average height of the buildings is 20 m), which probably modify flow field and create artifacts in the rotation angle and lead to turbulent flux estimates deviation. Given that Miyun and Helsinki have similar land use characteristics (many residential buildings (~18 m) are located at the west, northwest, and north of the Miyun tower), planar fit rotation was not adopted in this study to avoid possible data estimates errors. Moreover, when applying the planar fit rotation, it needs a long period (days to weeks) to estimate the differences between the anemometer alignment and the mean stream field for a given measurement site (Foken, 2008). While at Miyun, the rapid growth of crops will lead to the continuous change of underlying surface height and flow field, which make it difficult to meet the time conditions for planar fit method application. It is suggested to use the double coordinate method when canopy height and roughness change quickly, such as during the growing season in a crop field (LI-COR, 2019).
Secondly, the double coordinate method has good applicability and is widely used in the EC data processing of urban and suburban sites. To some extent, the data from different observation sites are more comparable with the same data processing method.
In summary, for the reasons mentioned above, the double coordinate method is selected to use in this study.

Vesala et al (2008) found that the planar fitted fluxes were generally 10% lower than

those obtained by double coordinate rotation. However, there is no specific standard or index to evaluate the fluxes of the two methods at present. Given most of the urban or suburban fluxes adopted the double coordinate rotation, it makes Miyun results comparable with those reports of research sits.

Foken, T.: Micrometeorology, Berlin: Springer Nature, 165-169. doi: 10.1007/978-3-642-25440-6, 2017.

LI-COR, Inc: EddyPro software instruction manual, 2-26pp, Version 7.0, 2019.

Vesala, T., Järvi, L., Launiainen, S., Sogachev, A., Rannik, Ü., Mammarella, I., Siivola, E., Keronen, P., Rinne, J., Riikonen, A., and Nikinmaa, E.: Surface-atmosphere interactions over complex urban terrain in Helsinki, Finland, Tellus. Ser. B. Chem. Phys. Meteor., 60, 188–199, doi:10.1111/j.1600-0889.2007.00312.x, 2008.

* l.101-103, From Table 1 I can't tell how much data are excluded for each reason...for example, how much data are excluded due to the LICOR poor-quality flag?  How much is excluded due to precip?  You should provide the reader with these details in Table 1...

 The amount of the deleted $Q_H$ and $Q_E$ data due to poor quality (N=1552 (6.6%) of $Q_H$, 1987 (8.5%) of $Q_E$), during rain and two hours after rain (N=615 (1.4%)) has been supplemented on line 109-110 of revised version. At the same time, Table 1 has been deleted from the revised manuscript because its' data availability is displayed too cumbersome.

* l.104.   I thought the EC sensor (ie, sonic) was installed pointing into the prevailing wind direction (see l.92), but now you are saying that it was installed based on the magnetic north?   As long as the sonic orientation is fixed for the entire project there shouldn't be any need to correct WD for changes in the magnetic declination...the boom/sonic direction should be measured relative to true north (if using a compass to do this, then the declination angle needs to be accounted for)....or, perhaps I don't understand the point of this sentence?

 EC sensor was installed pointing to the prevailing wind direction (90º), but this 90º is determined with a compass. The north direction determined by the compass is geomagnetic north, which has a declination angle with the geographical north. The north direction of land use/cover maps is geographical north.
 When the characteristics of turbulent flux in a certain direction are analyzed, the impact of land use in that direction on the turbulent values will be considered. At this time, there exists an angle deviation between the direction of the turbulence and that of the land use/cover map, so the magnetic declination needs to be corrected.

* l.108, how much radiation data was removed due to the radiation data being outside of "physically reasonable thresholds"?

 That radiation data beyond the physically reasonable threshold refers to those non-zero incoming and outgoing shortwave radiation data at night. These shortwave radiation data account for 5.2% of the available data. Given these data values are very small, basically no more than $1\times10^{-5}$, they did not been removed, but are forced to zero.

* l.110, "instrument failures". which instrument? the sonic? The IRGA? sometimes these failures are only for a day?

The data gap from January 14 to February 18, 2013, is due to CR3000 data-logger failure. Other short-term data gaps happened during the power cut caused by instrument inspection and maintenance. The data gap for one day was only a few hours of a power failure and did not last for the whole day.

To make the manuscript more rigorous, and also refer to the suggestion of another reviewer, the description for data gaps of a short time period has been deleted in the revised manuscript (Line 123).

* l.119, I did a google search for "ZQZ-TF, Aerospace Newsky Technology" and didn't find any information about this sensor. maybe it's easy to find it in Chinese, but not English? is it a sonic? prop-vane? since it might be a new sensor/model/company to the english-speaking world a few more details are needed...maybe provide a web link to more info?

ZQZ-TF is the model of wind sensor. Wind speed is captured using an anemometer with three lightweight, conical cups. Wind direction is captured via a wind vane.

It is easy to find the Chinese description of this instrument, such as through this website, https://max.book118.com/html/2019/0709/7033066031002040.shtm, but the English description is not found.

The website of Aerospace Newsky Technology CO., LTD is https://www.js1959.com, but I don't know why this website is currently inaccessible.

Since effective English information about the instrument and manufacturer cannot be provided at present, the web link is not added to the revised manuscript.

* l.122, what is the WUSH-BH? A data logger? Why do you need it to average to 1-min? Also, why not use the same averaging period as with the EC system (ie, 30-min)?

WUSH-BH is the model of the data logger.

The processing flow of observation data from meteorological stations is uniformly stipulated by China Meteorological Administration and cannot be changed at will.

When starting the data analysis of this paper, we can only download the hourly averaged data of Miyun meteorological station and cannot get the raw data to calculate 30 min mean values.

* l.135-136, Is there are reason "Normal" is capitalized? (seems to be capitalized throughout the entire paragraph/paper).

All "Normal" changed to "normal" in the revised manuscript.

* l.142-143, May seems very low/strange. Could this be a sensor problem? Does soil moisture corroborate the precip measurements? It also seems unlikely because the latent heat fluxes are fairly large in May 2013 (ie, Fig. 6f). It seems unlikely that Qe would increase if there was no rain that month...or perhaps you have an explanation for this? [ok, I see this discussed in Sect 3.5].

The precipitation in May is indeed very small. We have checked the precipitation repeatedly and queried the rainfall data of the automatic weather stations around the Miyun site, which are also small in May. It can be seen from Figure 9 (reordered as Figure 14 in the revised version) that soil

moisture under natural conditions has been decreasing in May due to small rainfall. As we discussed in Section 3.5, larger $Q_E$ in May could be attributed to supplement water supplied by cropland irrigation.

\* l.147, what is "existence hours"?

"existence hours" changed to "the start and end times" (Line 161).

\* l.160, if the buildings are 50.4 m and your measurement level is at 36m, then you are in the roughness sublayer and not the surface layer.

The building with a height of 50.4 m is located 180-185º south of the tower. For this direction, the observation height of 36 m is indeed in the rough sublayer rather than the constant flux layer. The turbulent heat flux in this direction represents the micro-scale energy exchange between earth and atmosphere rather than the local scale. However, the available flux data in this direction only account for 2.3% of the total data, which has little impact on the conclusions of this paper. Moreover, when energy fluxes and ratios are compared between farmland and buildings-dominated directions, the observed data in this direction did not participate in the calculation.

\* l.177, For "During the observation period" you should point out that these are from the quality-controlled statistics.

The sentence (Line 194-195) is changed to "During the observation period, after quality control (N=17142, 30 min periods), the stability is predominately unstable (40.9%) and stable (42.4%), with neutral conditions for 16.7% of the time.

\* l.195, The two papers cited (Wilson, et al 2002; Foken et al 2008) are rather old. There has been a lot of work in this area since then, you should provide a more recent reference and summary of recent work done in this area...

These two references are cited to explain that the underestimation of $Q_H$ and $Q_E$ is one of the reasons that $\Delta Q_{S,res}$ is the upper limit of $\Delta Q_S$. It is not the focus of this paper to discuss the energy imbalance of EC observation. Given these two references are quite classic, a cite to Wilson et al. (2002) and Foken et al. (2008) is still retained.

\* l.205, what does, "Daytime and daily mean fluxes of net all-wave radiation, sensible heat flux and latent heat flux are estimated based on monthly mean diurnal patterns." mean? How do you get daily mean fluxes from the mean monthly diurnal pattern? I don't understand this statement.

This sentence wants to express that because the missing data are not gap-filled, the monthly (seasonal) daytime and daily mean fluxes cannot be obtained by directly averaging the available 30 min data of the month (season). It needs to calculate the mean value of every 30 min within 24 hours of the month (season) firstly (i.e. the mean value of 00:00, 00:30, 01:00, 01:30, …… 23:00, 23:30 in a day), and then average the daytime ($K_\downarrow > 5$ W m$^{-2}$) or daily (24 h) period. The monthly (seasonal) daytime or daily mean ratios (e.g. albedo, Bowen ratio, etc.) are estimated based on the monthly (seasonal) daytime or daily mean fluxes.

This sentence has been rewritten as follows (Line 223-229 in the revised manuscript):
Missing data are not gap-filled. The mean values of radiation and turbulent fluxes for each half-hour during a day in the month (season) are first calculated to get their mean diurnal patterns. Then

the daytime ($K_\downarrow > 5$ W m$^{-2}$) or daily (24 h) mean values are averaged from corresponding periods within the mean diurnal patterns. The daytime (daily) mean ratios are the ratios of daytime (daily) mean values of corresponding radiation and energy fluxes.

* l.209, "At the MY site, all radiation fluxes vary seasonally". This statement is true for any location on planet Earth. I don't think you need to inform readers of this...

The sentence 'all radiation fluxes vary seasonally" has been deleted in the revised manuscript (Line 232).

* l.220, rewrite, "...causing albedo becomes increases..."

Done. Changed to "becomes an increase to 0.6 and then a decrease with days since snowfall" (Line 245-246).

* l.238, rewrite, "...small impacts presence of snow...".

This sentence (Line 271-273) has been deleted in revised manuscript.

* l.263, "Hence, the MY values appear to be reasonable.". Can you explain why there is such a large variation in Qf among these different locations? What you wrote doesn't lead me to conclude that the MY Qf values are "reasonable"...

A few sentences of the $Q_F$ comparison between Miyun and other suburban sites (Montreal, Swindon) in the world has been added to the revised manuscript (L309-325). The newly added content explains the rationality of Miyun $Q_F$ value and the differences with those values of other suburban sites by comparing population density, temperature, heating/cooling demand, etc.

Text added (L309-325):
The median $Q_F$ vary between 5 and 39 W m$^{-2}$ diurnally across all of the months (Fig. 6). Heat released from buildings ($Q_{F,B}$) dominated $Q_F$, with its diurnal median value ranging from 4.6 to 36.4 W m$^{-2}$ and accounting for 85-95% of $Q_F$ (Fig. S2a). Contribution of vehicle emission ($Q_{F,V}$) is small (1-9% of $Q_F$). The maximum $Q_{F,V}$ value during the morning rush hour (8:00-9:00) is only 1.5 W m$^{-2}$, since the house is usually close to the working place or school and residents do not rely on cars for daily travel or commuting at MY (Fig. S2b). The diurnal variation of human metabolism ($Q_{F,M}$) is the constant within a year because of the fixed population density (5657 people km$^{-2}$ in 2012 and 5702 people km$^{-2}$ in 2013). Like other studies, $Q_{F,M}$ is small, with values ranging between 0.4-1 W m$^{-2}$ and contributions being 5-8%.

The fluxes are larger in summer and winter associated with cooling and heating needs, respectively. Our $Q_{F,B}$ values are larger than estimated in other suburban sites, such as Montreal during winter (Bergeron and Strachan, 2012) and Swindon (Ward et al., 2013), even the air temperature in Montreal is lower in winter. The reason is that Miyun has a greater mean building height (13.1 m) and building cover (21%). Moreover, buildings include hospital and office buildings, which usually have greater energy consumption and emissions than residential buildings. However, our $Q_{F,V}$ values are lower than those reported in Montreal and Swindon, due to the small number of motor vehicles and various travel modes at Miyun.

[Figure]

**Figure 6:** Monthly anthropogenic heat flux ($Q_F$) at Miyun (September 2012 to December 2013) calculated as described in Section 2.2: (a) mean total for 24 h, (b) median diurnal patterns with inter-quartile range (IQR) (shading).

[Figure]

**Figure S2:** Monthly median diurnal pattern (points) with inter-quartile ranges (shading) of (a) anthropogenic heat flux ($Q_F$) and heat released from buildings ($Q_{F,B}$), (b) heat released from traffic ($Q_{F,V}$) and human metabolism ($Q_{F,M}$).

\* l.276, "...not including all components of heat storage flux, such as biomass heat storage". Why don't you include the biomass storage?

To estimate biomass heat storage, it needs observation data of canopy temperature and leaf area index, and the values of specific leaf weight (mass of dry leaves per square meter of the leaf) and the water content on a wet master basis. However, we don't have these data and cannot calculate it.

\* l.323, not only evaporation, but also transpiration (assuming there are crops/plants).

The word "transpiration" has been added to the revised manuscript (Line 439).

\* l.342-345, not sure you need to re-state this since it's already discussed in the previous paragraph...

Restatement has been deleted from the revised manuscript.

Sentences (Line 461-464 in the revised version) has been modified to "Monthly totals of $Q_E$ are between 0.32 and 7.16 MJ d$^{-1}$ m$^{-2}$ (Fig. 9e), with corresponding maximum and minimum monthly totals occur in two seasons (summer - July and winter - December 2013, respectively)."

* l.425-427, Fig. 9 is very helpful. That irrigation provides larger Qe is logical and having the soil moisture measurements to show this is very helpful/useful.

Figure 9 (reordered as figure 14 in the revised version) is slightly adjusted to show the soil moisture in 2012 and 2013 more clearly.

[Figure]

**Figure 14:** Gravimetric soil moisture (%) at a depth of 0.1 m measured on the 8th, 18th and 28th of each month at the MY site from 18 March to 18 November, 2013 under natural condition and irrigated cropland. The shading is May and July 2013, respectively.

* l.430-432, Fig. 11, these results could also be influenced by different magnitudes in the storage terms (which are not accounted for) for the land surface types.

We agree with the reviewer's comments. The heat storage terms ($\Delta Q_S$) differ in land use/cover types. So, the $\Delta Q_S$ of different land use/cover types have varying degrees of influence on their respective $Q_H$ and $Q_E$, and energy flux ratios ($Q_H/Q_\downarrow$ and $Q_E/Q_\downarrow$). However, given $\Delta Q_S$ is calculated as a residual term of the energy balance equation and its uncertainty is relatively large, it is difficult to accurately quantify the impact of $\Delta Q_S$ on the $Q_H/Q_\downarrow$ and $Q_E/Q_\downarrow$ of land use/cover dominated by impervious surfaces and farmland, respectively. Thus, we do not discuss it here.

* l.432-435, Fig. 12, this is an interesting result and the interpretation seems plausible. However, I'm not sure I fully understand what is shown in Fig. 12...for example, what does a frequency of < 60 (blue box) mean?

Figure 12 (reordered as Figure 17 in the revised version) in the manuscript shows the frequency distribution of Bowen ratios ($\beta$) in different ranges with the time variation since the last rainfall. The different color blocks in the legend represent frequency range (F), i.e. "F=0, 0 < F <5%, 5≤F< 10%, …… 55≤F < 60 %, 60 ≤ F < 65%".

Taking the frequency of Bowen ratio at farmland-dominated direction in the 14th hour since the last rain as an example (Table rev1), the frequency in the range of 0 < β ≤ 0.5 is 56%, so it is represented by a dodger blue block of 55 ≤ F < 60% (abbreviated as "< 60") in the Figure 12b (Figure 17b in revised version).

Table rev1. Frequency of Bowen ratio at the farmland-dominated direction in the 14th hour since the last rain.

| Bowen ratio range | Frequency (%) |
| --- | --- |
| ≤ -1 | 2 |
| (-1, -0.5] | 0 |

| | |
|---|---|
| (-0.5, 0] | 15 |
| **(0, 0.5]** | **56** |
| (0.5, 1] | 21 |
| (1, 1.5] | 6 |
| (1.5, 2] | 0 |
| (2, 2.5] | 0 |
| (2.5, 3] | 0 |
| (3, 3.5] | 0 |
| (3.5, 4] | 0 |
| (4, 4.5] | 0 |
| (4.5, 5] | 0 |
| (5, 5.5] | 0 |
| (5.5, 6] | 0 |
| (6, 6.5] | 0 |
| (6.5, 7] | 0 |
| (7, 7.5] | 0 |
| (7.5, 8] | 0 |
| > 8 | 0 |

[Figure]

**Figure 17:** Frequency (F, %) of Bowen ratio $\beta$ values by time since last rainfall for (a) building-dominant (210-360°) and (b) farmland-dominant (30-150°) directions.

References:

========================

Leuning, R., van Gorsel, E., Massman, W. J., and Isaac, P. R.: Reflections on the surface energy imbalance problem, Agr. Forest Meteorol., 156, 65-74, doi: 10.1016/j.agrformet.2011.12.002, 2012.

Swenson, S. C., Burns, S., and D. Lawrence: The impact of biomass heat storage on the canopy energy balance and atmospheric stability in the Community Land Model, Journal of Advances in Modeling Earth Systems, 11, 83-98, doi:10.1029/2018MS001476, 2019

Thank you very much for the recommended references. We have read them carefully but did not cite them, as the manuscript did not focus on the EC energy balance.